# SIGMA: SEMANTICALLY INFORMATIVE PRE-TRAINING FOR SKELETON-BASED SIGN LANGUAGE UNDERSTANDING

## ABSTRACT

Pre-training has proven effective for learning transferable features in sign language understanding (SLU) tasks. Recently, skeleton-based methods have gained increasing attention because they can robustly handle variations in subjects and backgrounds without being affected by appearance or environmental factors. Current SLU methods continue to face three key limitations: 1) weak semantic grounding, as models often capture low-level motion patterns from skeletal data but struggle to relate them to linguistic meaning; 2) imbalance between local details and global context, with models either focusing too narrowly on fine-grained cues or overlooking them for broader context; and 3) inefficient cross-modal learning, as constructing semantically aligned representations across modalities remains difficult. To address these, we propose Sigma, a unified skeleton-based SLU framework featuring: 1) a sign-aware early fusion mechanism that facilitates deep interaction between visual and textual modalities, enriching visual features with linguistic context; 2) a hierarchical alignment learning strategy that jointly maximises agreements across different levels of paired features from different modalities, effectively capturing both fine-grained details and high-level semantic relationships; and 3) a unified pre-training framework that combines contrastive learning, text matching and language modelling to promote semantic consistency and generalisation. **Sigma** achieves new state-of-the-art results on isolated sign language recognition, continuous sign language recognition, and gloss-free sign language translation on multiple benchmarks spanning different sign and spoken languages, demonstrating the impact of semantically informative pre-training and the effectiveness of skeletal data as a stand-alone solution for SLU. We will release the code upon acceptance.

## 1 INTRODUCTION

Sign languages (**SL**s) are the primary means of communication for around 70 million people with hearing or speech impairments, spanning more than 200 SLs worldwide (WHO, 2025; WFD, 2025). SLs remain challenging for the general public to master due to the global diversity and complex structure, which encompasses rapid and intricate hand gestures, body postures, as well as facial expressions. The ultimate goal of sign language understanding (**SLU**) is to comprehend SLs at the levels of words, phrases, as well as sentences anytime and anywhere for the impaired community, enabling barrier-free communication for them. Achieving this goal requires the development of models capable of interpreting these visual signals in alignment with the unique linguistic structure of SLs. SLU typically comprises three core tasks: isolated sign language recognition (**ISLR**), which recognises sign glosses [1] (Hu et al., 2021a; 2023b; Pu et al., 2024); continuous sign language recognition (**CSLR**), which aligns unsegmented sign sequences with sign glosses (Hu et al., 2021a; Zuo & Mak, 2022; Fu et al., 2025); and sign language translation (**SLT**), which converts sign sequences into sentences (Zhou et al., 2021a; 2023; Fu et al., 2025). These tasks demand both fine-grained visual recognition and strong contextual understanding.

Recently, SLU research has progressively shifted from fully supervised learning toward the development of effective pre-training paradigms, commonly referred to as sign language pre-training (**SLP**)

---

[1] A sign gloss is a textual label that represents the meaning of a sign sequence using a word or phrase.

(Zhou et al., 2023; Hu et al., 2023b; Zhou et al., 2024; Li et al., 2025; Fu et al., 2025). These methods present a promising direction by enabling models to learn transferable representations directly from SL data, thereby significantly reducing the reliance on manual annotations, such as gloss annotations, temporal boundaries or clip-level supervision. By capturing structural and temporal regularities during the pre-training stage, models gain generalizable knowledge that accelerates convergence and enhances performance on a wide range of downstream SLU tasks. Consequently, SLPTserves as a foundational step toward building unified and scalable SLU frameworks. Despite their potential, current SLP-based SLU methods continue to face significant limitations.

First, the **lack of semantic grounding** in visual representations remains a major challenge in advancing SLU. While dense geometric features in skeletal data, such as hand trajectories, body movements, and facial expressions, provide important visual cues, they often carry limited linguistic meaning. Most existing skeleton-based SLU methods focus on capturing these low-level patterns from skeletal data, treating SL primarily as a visual signal and paying little attention to the underlying linguistic structure (Hu et al., 2021a; 2023b; Zhao et al., 2024b; Pu et al., 2024). Although such models may capture low-level motion patterns, they struggle to model the relationship between these geometric features and their intended semantic roles. This disconnect weakens the ability of models to produce accurate and meaningful interpretations. Addressing this issue requires enriching visual features with semantic grounding, allowing the model to understand both the appearance and the purpose of each gesture. Doing so helps bridge the gap between visual representation and language understanding, making the model capable of supporting accurate recognition and fluent translation.

Second, the **imbalance between local-global feature modelling** remains a persistent challenge in SLU, which inherently spans both recognition and translation tasks. Accurately distinguishing subtle variations in SL gestures requires capturing fine-grained local motion patterns, while achieving coherent understanding necessitates preserving high-level global semantics. Balancing these two levels of representation is inherently difficult but critical (Liu et al., 2013). Global semantic modelling plays a key role in resolving ambiguities between visually similar SL gestures, particularly in continuous streams where the boundaries of sign glosses are unclear and context determines meaning. In such cases, local features alone are inadequate. Conversely, precise local detail extraction is equally vital, as small variations in hand gestures, body postures, or facial expression can dramatically alter meaning and grammatical structure. Even minor changes in motion intensity may shift interpretation and degrade translation quality (Camgoz et al., 2018). Therefore, robust SLU demands a mechanism that jointly models both local and global features in a balanced manner.

Third, **inefficient cross-modal representation learning** remains a critical bottleneck for advancing SLU. Compared to traditional video understanding, SLU from RGB videos is more challenging because gestures and facial expressions are more intricate or rapid than general human actions or scene changes (Hu et al., 2021a; 2023b). Constructing structured, semantically aligned representations from raw visual streams is difficult, as models are easily distracted by background details or appearance variations rather than focusing on the linguistic cues that carry meaning (Hu et al., 2021a; 2023b; Pu et al., 2024). This inefficiency weakens the alignment between dynamic gestures and textual semantics while imposing heavy computational and storage costs, ultimately limiting the scalability of SLU and slowing progress in SL production and generation. Skeletal data offers a promising alternative to RGB videos (Hu et al., 2021a; 2023b; Pu et al., 2024). It intentionally prioritises the essential spatial-temporal dynamics of SL, which are the core semantic carriers in SLU. Modest estimation variances in skeletal data can improve generalisation across diverse real-world motion patterns, and by abstracting away visual noise such as lighting, background clutter, and appearance biases, skeletal representations provide cleaner, more relevant inputs with stronger privacy guarantees.

Collectively, there is a need for an approach that enhances meaningful semantic grounding, promotes balanced feature modelling, and supports effective cross-modal representation learning for skeleton-based SLU. Visual illustrations of our motivation are provided in Appendix A. To overcome these limitations, this paper proposes the following solutions:

- We introduce **a sign-aware early fusion mechanism** that enables bidirectional interaction between visual and textual features during the encoding stage. This encourages the model to learn semantically enriched visual representations, improving modality alignment and deepening contextual comprehension.
- We propose **a hierarchical alignment learning strategy**, which learns representations by maximising agreement across modalities. This enables the model to capture fine-grained

visual cues and high-level semantic structures, supporting accurate recognition and fluent translation.

- We design **a skeleton-based unified cross-modal pre-training framework** that facilitates efficient and flexible representation learning across multiple tasks. By jointly optimising contrastive learning, text matching, and language modelling within a shared space, the framework improves semantic alignment as well as boosts transferability and generalisation across diverse downstream SLU tasks.

## 2 RELATED WORK

### 2.1 SEMANTICALLY INFORMATIVE VISUAL FEATURE

Learning semantically informative visual features from SL sequences is crucial for understanding. This is particularly important in resolving the representation density problem, where visually similar SL gestures, differing only slightly in motion or expression, tend to cluster closely in feature space (Ye et al., 2024). Incorporating linguistic and contextual cues into visual representations helps mitigate feature overlap and enables the model to learn more separable and discriminative features. This can lead to improved performance in both recognition and translation tasks, especially in cases where subtle visual differences correspond to distinct meanings. Prior works such as TSPNet (Li et al., 2020b), GLE-Net (Hu et al., 2021c), HST-GNN (Kan et al., 2022) and SignCL (Ye et al., 2024) have made progress in temporal modelling, global context extraction, and multi-perspective graph-based reasoning. However, learning and embedding semantically rich visual features in a way that generalises across tasks remains an open challenge in advancing SLU.

### 2.2 SIGN LANGUAGE UNDERSTANDING

SLU has been widely studied through task-specific methods. Prior works for ISLR have applied spatial-temporal modelling to improve accuracy (Hu et al., 2021b; Li et al., 2020c; Zuo et al., 2023). Recent models for CSLR address co-articulation and gloss boundary ambiguity using CTC-based or sequence-to-sequence frameworks (Min et al., 2021; Hu et al., 2023e; Jiao et al., 2023). For SLT, gloss-based approaches rely on intermediate gloss annotations (Camgoz et al., 2020; Zhou et al., 2021b), while emerging gloss-free methods adopt pre-training and large language models to reduce annotation requirements and improve generalisation (Zhou et al., 2023; Wong et al., 2024; Gong et al., 2024). In this study, we focus on the gloss-free SLT paradigm and aim to enhance its effectiveness by learning semantically rich visual representations aligned with textual outputs. In contrast to prior task-specific methods, we propose a unified framework capable of performing all aforementioned SLU tasks. A central design motivation lies in the differing representational needs across tasks of recognition and translation.

### 2.3 SIGN LANGUAGE PRE-TRAINING

SLPTmethods employ pretext tasks to learn useful representations from SL data, improving downstream performance. Self-supervised models like SignBERT (Hu et al., 2021a; 2023b) use masking and reconstruction to capture visual patterns from unlabeled videos but often lack sufficient sematic grounding. To address this, MMTLB (Chen et al., 2022a) introduces multi-task training across sign-to-gloss, gloss-to-text, and sign-to-text objectives, while GFSLT-VLP (Zhou et al., 2023) uses contrastive learning for sign-text alignment. More recent efforts, including MSLU (Zhou et al., 2024) and C$^2$RL (Zuo & Mak, 2022), incorporate keypoint reconstruction and language modelling to enhance semantic representation. Despite these advances, most approaches remain task-specific, limiting scalability. Moreover, they often struggle to balance modality-specific encoding with effective cross-modal transfer, both of which are essential for developing unified and generalisable SLU systems.

## 3 METHOD

Sigma consists of two stages: pre-training and fine-tuning (Figure 1). In pre-training, we introduce sign-aware early fusion (**SignEF**) for deep bidirectional cross-modal interaction, hierarchical align-

ment learning (**HAL**) for multi-level semantic alignment to capture both coarse and fine-grained semantic correspondences, and a sign-grounded text (**SGT**) encoder jointly trained with text matching and language modelling to enhance semantic consistency and linguistic fluency. The sign encoder is fully transferred, and the SGT encoder is reused in the unified fine-tuning, enabling consistent and efficient adaptation across SLU tasks, including ISLR, CSLR, and SLT.

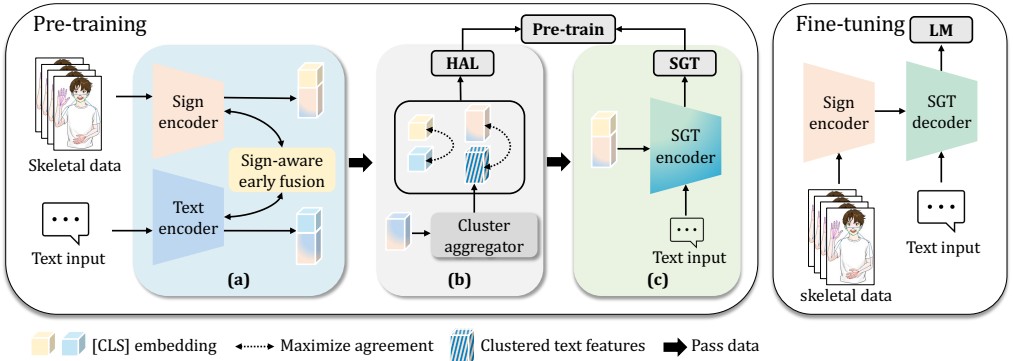

Figure 1: Overview of Sigma. (a) SignEF enhances visual-linguistic alignment by injecting cross-modal features into sign and text encoders. (b) HAL is used to maximise global and local cluster agreement. (c) SGT encoder jointly optimises sign-text matching and language modelling. During fine-tuning, both the sign and SGT encoders are reused across SLU tasks.

## 3.1 PRELIMINARIES

We use paired skeletal data and their corresponding text(s) for both the pre-training and fine-tuning stages. The text(s) are tokenised before being fed into the text encoder. The skeletal data are 2D keypoints estimated from SL videos using RTM-Pose (Jiang et al., 2023). Part-specific ST-GCNs (Yan et al., 2018) are used to model both joint interdependencies and motion dynamics. Following these ST-GCNs, the raw skeletal input $S_p^{\text{raw}} \in \mathbb{R}^{L \times N_p \times C}$ is projected into a compact feature $S_p \in \mathbb{R}^{L \times D}$, where $L$ is the sequence length, $N_p$ is the number of keypoints in a group that $p \in \{lh, rh, b, f\}$ (left hand, right hand, body, and face), $C$ is the visual input dimension, and $D$ is the projected dimension. The sign encoder input $S \in \mathbb{R}^{L \times 4D}$ is formed by concatenating features from all groups and serves as the visual input for the two-stage training of Sigma.

## 3.2 SIGN LANGUAGE PRE-TRAINING

We initialise Sigma with pre-trained mT5 (Xue et al., 2021; Li et al., 2025) to leverage large-scale text corpora for stronger visual-linguistic alignment.

### 3.2.1 SIGN-AWARE EARLY FUSION MECHANISM

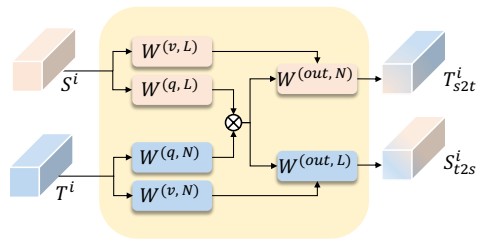

Figure 2: SignEF promotes progressive visual-linguistic interaction with parameters $W^{(x,L)}, W^{(x,N)} : x \in \{q, v, \text{out}\}$, analogous to query, value, and output projections by (Vaswani et al., 2017).

A key challenge in skeleton-based SLU is aligning geometric gesture features with textual semantics. Inspired by (Vaswani et al., 2017; Li et al., 2022b), we propose SignEF, which enriches SL representations by introducing cross-modal interaction at the encoding stage, fostering more expressive and semantically aligned features. Specifically, SignEF deploys cross-attention and injects textual cues into visual encoding layers, enabling the model to perform deep and structured representation learning across modalities.

Let $S^i$ and $T^i$ denote the visual and textual features from the $i$-th layers of the sign and text

encoders. Their fusion outputs, $S_{t2s}^i$ (text-to-sign) and $T_{s2t}^i$ (sign-to-text), are fed back into the encoders, with early fusion applied at the last few layers of the encoders. The process is defined as:

$$
\begin{aligned}
S_{t2s}^i, T_{s2t}^i &= SignEF(S^i, T^i) \\
X^{i+1} &= Mo\text{-}Encoder_{i+1}\left(X^i + X_{\bar{m}2m}^i\right)
\end{aligned}
\tag{1}
$$

where $X$ denotes either sign ($S$) or text ($T$) features, with $Mo \in Sign, Text$ indicating the target modality and $m \in s, t$ the source. The SignEF module lets one modality attend to the other, computing cross-modal context features. As shown in Figure 2, attention heads share parameters across the final SGT layers. This parameter-sharing design prompts fine-grained visual–linguistic alignment while keeping the model efficient.

### 3.2.2 HIERARCHICAL ALIGNMENT LEARNING

Balancing local and global feature modelling is essential for SLU, where recognition and translation require attention to both detailed and holistic semantics. Inspired by contrastive learning (Chen et al., 2020; Radford et al., 2021; Li et al., 2022a; Hou et al., 2024), we introduce SignEF as a core strategy for pre-training. SignEF maximises agreement between sign-text pairs at both the global and local cluster levels. The similarity is computed as:

$$
\mathbf{M}_{s2t}^x = sim(S_f, T_f; \phi) = \begin{cases} g_s(s_{cls})^\top g_t(t_{cls}), \text{ if } x = g & (2.1) \\ \sum_{i=1}^n max_{j \in \{1,\ldots,k\}}\left(g_s(s_i)^\top g_t(c_j)\right), \text{ if } x = l & (2.2) \end{cases}
\tag{2}
$$

where $f$ denotes features, $l$ indicates local, and $g$ means global. Globally, we align the class-token representations $s_{cls}$ and $t_{cls}$ from the sign and text encoders. These class tokens are projected into a shared embedding space using projection heads $g_s$ and $g_t$, enabling the model to capture coarse-grained semantic relationships across modalities. A cluster aggregator (see Figure 3) compresses text tokens into $k$ clusters, which serve as compact semantic units to correspond to the $n$ sign tokens. To focus on cross-modality alignment, we compute the maximum similarity between each sign token $s_i$ and all text clusters $c_j$. Locally, SignEF enhances fine-grained interaction by computing local cluster-wise similarity between sign features and clustered text features.

---

**Algorithm 1** Cluster-wise sign-to-text similarity (see Figure 9 in the Appendix I.1 for visualisation)

---

1: **Input:** Sign tokens $\{S_b \in \mathbb{R}^{N \times D}\}_{b=1}^B$, Textual clusters $C \in \mathbb{R}^{B \times K \times D}$
2: **Output:** Cluster-wise sign-to-text similarity $\mathbf{M}_{s2t}^l$
3: Initialise $\mathbf{M}_{s2t}^l \leftarrow \mathbf{0}^{B \times B}$
4: **for** $i = 1$ **to** $B$ **do**
5:     $M \leftarrow S_b C^\top$             ▷ Compute cosine similarity matrix $M \in \mathbb{R}^{B \times N_b \times K}$
6:     $R \leftarrow max(M, \dim = 3)$             ▷ Row-wise operation $R \in \mathbb{R}^{B \times N_b}$
7:     $w \leftarrow softmax(R)$
8:     score $\leftarrow \sum_{\dim=2}(w \odot R)$             ▷ Local-level scoring $score \in \mathbb{R}^B$
9:     $\mathbf{M}_{s2t}^l[b] \leftarrow$ score
10: **end for**

---

Glosses serve as simplified representations of SL segments in continuous video, and the extra supervision they provide has significantly improved SLU performance. However, they come with many limitations (see Appendix H). Our aggregator promotes hierarchical alignment by approximating gloss-like groupings through local feature clustering. It groups subword-level textual tokens into semantically meaningful units. For instance, "curiosity" is split into "curios" and "ity," and "背包" into "背" and "包" by tokenizers. Both are expressed as continuous sign sequences, while each subword could be intended to align with distinct visual segments. Our aggregator dynamically merges them into more concrete phrase-level units, with the number of clusters adaptively determined by sentence structure and bounded by sentence lengths. This enables our model to preserve compositional semantics and reduce alignment errors by preventing disjoint mappings of continuous signs.

We express the computation of the local cluster-wise similarity at Algorithm 1. For brevity, Algorithm 1 only presents the local sign-to-text similarity $\mathbf{M}_{s2t}^l$, and the local text-to-sign similarity is computed

analogously, with the positions of sign tokens and textual clusters exchanged. We experiment with several different row-wise operations and local-level scoring methods in Appendix I.1 to provide a better understanding of the design choice.

SignEF is adopted to align different levels of paired features from different modalities. This dual-level strategy encourages semantically meaningful and discriminative cross-modal representations. The global and local losses are computed as follows:

$$\mathcal{L}^{\phi}(S_f, T_f) = \frac{1}{2}\left(L_{s2t}^{\phi}(S_f, T_f) + L_{t2s}^{\phi}(T_f, S_f)\right)$$

$$L_{s2t}^{\phi}(S_f, T_f) = -\frac{1}{b}\sum_{i=1}^{b}\log\frac{\exp\left(\text{sim}(S_f^i, T_f^i; \phi)/\tau_\phi\right)}{\sum_{j=1}^{b}\exp\left(\text{sim}(S_f^i, T_f^j; \phi)/\tau_\phi\right)} \tag{3}$$

For each sign-text feature pair $(S_f^i, T_f^i)$ in a batch $b$, we compute the bidirectional global contrastive loss $\mathcal{L}^{\phi}$ with temperature-scaled similarity $\tau\phi$ controlled by parameters $\phi$ (as shown Equation 3). The goal is to maximise similarity for matched pairs and minimise it for mismatches $(S_f^i, T_f^j)$, $j \neq i$. We show the sign-to-text loss $L_{s2t}^{\phi}$ explicitly; the text-to-sign loss $L_{t2s}^{\phi}$ is omitted for brevity, as it is defined in the same manner as $L_{s2t}^{\phi}$, with the roles of text and sign features reversed. The local contrastive loss follows the same structure but omits temperature scaling to emphasise sharper fine-grained alignment.

To balance both alignment levels, we introduce a parameter $\alpha \in [0, 1]$, and define the SignEF loss as:

$$\mathcal{L}_{HAL} = (1-\alpha)\mathcal{L}_{global}^{\phi}(S_f, T_f) + \alpha\mathcal{L}_{local}^{\phi}(S_f, T_f) \tag{4}$$

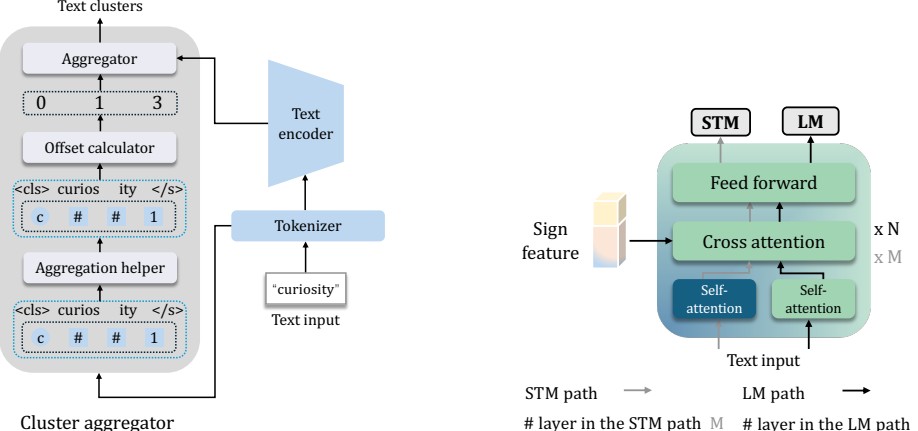

Figure 3: The overview of the cluster aggregator module. It converts sub-word token embeddings into cluster-level representations by grouping tokens, mapping them with offset indices, and aggregating hidden features for semantic alignment with visual inputs (See Appendix D for details).

Figure 4: The architecture of the SGT encoder, which consists of two paths: the **STM** path injects sign features via cross-attention for semantic alignment, and the LM path preserves linguistic fluency through standard transformer layers (See Appendix E for details).

### 3.2.3 SIGN-GROUNDED TEXT MATCHING AND LANGUAGE MODELLING

To improve training efficiency and foster deeper cross-modal understanding, we propose an SGT encoder, inspired by (Chen et al., 2020; Radford et al., 2021; Li et al., 2021; 2022a). It unifies sign-text matching (STM) and language modelling (LM), supporting dynamic alignment of visual and linguistic features within a single framework. A task-specific token guides the model to produce multimodal embeddings. A lightweight *STM* head, trained with binary cross-entropy, judges sign-text

alignment. In parallel, the encoder autoregressively generates text with masked self-attention, with a cross-entropy *LM* loss enhancing language structure and semantics. To balance the synergy between matching and generation, we define a composite SGT loss:

$$\mathcal{L}_{SGT} = (1 - \beta)\mathcal{L}_{STM}(S_f, T) + \beta\mathcal{L}_{LM}(S_f, T), \tag{5}$$

where $\beta \in [0, 1]$ controls task emphasis. Matching enhances visual grounding, while generation regularises semantic coherence.

The pre-training objective integrates $\mathcal{L}_{HAL}$ and $\mathcal{L}_{SGT}$, defined as $\mathcal{L}_{pre\text{-}train} = \mathcal{L}_{HAL} + \mathcal{L}_{SGT}$.

### 3.3 Sign language fine-tuning

A unified architecture is designed for all the downstream SLU tasks, casting ISLR, CSLR, and SLT as conditional language modelling. The fine-tuning objective is defined as $\mathcal{L}_{task} = \mathcal{L}_{LM}(T_{out}, T_{task})$, where $T_{\text{out}}$ is the prediction, $T_{task}$ is the ground truth, and $task \in \{\text{ISLR, CSLR, SLT}\}$, with $T_{task}$ as a gloss (ISLR), a gloss sequence (CSLR), or a sentence (SLT).

## 4 Experiment

**Datasets**. We evaluate Sigma on a diverse set of benchmarks spanning different sign and spoken languages. WLASL2000 (Li et al., 2020a) is used for ISLR evaluation, CSL-Daily (Zhou et al., 2021a) serves as the benchmark both for CSLR and SLT. How2Sign (Duarte et al., 2021) and OpenASL (Shi et al., 2022) datasets are used for SLT evaluation (check Table 14 for their statistics and information).

**Evaluation metrics**. Following prior works, we report per-class (P-C) and per-instance (P-I) Top-1 accuracy for ISLR, word error rate (WER) for CSLR, and BLEU & ROUGE-L scores for SLT. For brevity, we denote BLEU-1, BLEU-4, and ROUGE-L as B@1, B@4, and R@L in the tables of the following sections.

Table 1: Training settings across tasks.

| Settings | ISLR | CSLR | SLT |
|---|---|---|---|
| optimiser | | AdamW | |
| Weight decay | | 1.00E-03 | |
| optimiser momentum | $\beta_1, \beta_2 = 0.9, 0.999$ | | |
| Learning rate schedule | | Cosine decay | |
| *Pre-training* | | | |
| Training epochs | 10 | 15 | 25 |
| Batch size | | 16 | |
| Learning rate | | 1.00E-06 | |
| *Fine-tuning* | | | |
| Training epochs | 10 | 15 | 15 |
| Batch size | | 8 | |
| Learning rate | 1.00E-07 | | 1.00E-06 |

**Training details**. The training settings are empirically configured and listed in Table 1.

### 4.1 Impact of sign-aware early fusion

To evaluate the impact of our SignEF, we vary the number of fusion layers that integrate visual and textual features within the encoders. As shown in Table 2, SignEF consistently improves performance, though the gains are not strictly linear. In CSLR and SLT, we observe a fluctuating trend, with alternating improvements and dips. Notably, applying two fusion layers yields the best results on CSL-Daily (highlighted in bold in Table 2), indicating that early fusion helps the model form stronger semantic dependencies, enhancing sequence alignment and translation quality. In contrast, ISLR benefits from deeper fusion, with performance peaking at five layers (highlighted in bold in Table 2), reflecting the need for precise modelling of fine-grained spatial and motion cues. These findings suggest that light fusion is beneficial for context-sensitive tasks like CSLR and SLT, while deeper fusion better supports visually intensive tasks such as ISLR. Overall, SignEF proves effective in adapting the depth of fusion to the needs of each SLU task.

### 4.2 Impact of local-global feature balancing

We introduce the trade-off parameter $\alpha$ in Equation 4 to balance the contributions of local-global feature learning. As shown in Table 3, setting $\alpha = 0.5$ consistently delivers the best results across ISLR, CSLR, and SLT tasks. This suggests that giving equal attention to fine-grained features such as motion and handshape, along with high-level semantics like context and meaning, leads to better sign-to-text mappings. Compared to SignEF, the performance fluctuation caused by different $\alpha$ values is relatively small for ISLR, more pronounced for CSLR, and moderate for SLT. When $\alpha$ shifts too far

in favor of either local features (values above 0.5) or global features (values below 0.5), we observe a decline in model performance, especially in CSLR and SLT, which rely heavily on contextual understanding. These findings highlight the importance of jointly modelling both detailed visual information and broader semantic structures to capture the complexity of SL.

Table 2: Impact of SignEF.

| Layers | WLASL2000 | | CSL-Daily | | | |
|---|---|---|---|---|---|---|
| | ISLR | | CSLR | | SLT | |
| | P-I↑ | P-C↑ | WER↓ | B@1↑ | B@4↑ | R@L↑ |
| 1 | 63.17 | 60.77 | 26.58 | 56.79 | 27.50 | 56.64 |
| **2** | 63.79 | 61.40 | **26.12** | **56.83** | **28.24** | **58.04** |
| 3 | 64.22 | 62.09 | 26.53 | 56.72 | 27.93 | 57.36 |
| 4 | 63.97 | 61.70 | 26.40 | 56.80 | 27.98 | 57.86 |
| **5** | **64.40** | **62.32** | 26.69 | 56.79 | 28.09 | 57.56 |

Table 3: Local-global feature balancing.

| Alpha | WLASL2000 | | CSL-Daily | | | |
|---|---|---|---|---|---|---|
| | ISLR | | CSLR | | SLT | |
| | P-I↑ | P-C↑ | WER↓ | B@1↑ | B@4↑ | R@L↑ |
| 0.2 | 64.14 | 62.14 | 26.52 | 56.11 | 27.99 | 57.89 |
| 0.4 | 64.30 | 62.24 | 26.55 | 56.65 | 28.12 | 57.82 |
| **0.5** | **64.40** | **62.32** | **26.12** | **56.83** | **28.24** | **58.04** |
| 0.6 | 64.14 | 62.09 | 27.05 | 56.82 | 27.99 | 58.10 |
| 0.8 | 64.14 | 62.08 | 26.65 | 56.21 | 27.66 | 57.65 |

## 4.3 TRADE-OFF ANALYSIS BETWEEN TEXT-MATCHING AND LANGUAGE MODELLING

The beta value $\beta$ in Equation 5 controls the relative contributions of matching and language modelling. Table 4 shows performance across SLU tasks under different $\beta$ values. Compared to the trade-off evaluated in Section 4.2, ISLR is more sensitive to the trade-off between text-matching & language modelling than CSLR and SLT. This indicates that different tasks respond differently to the balance between discriminative and generative learning. We observe that the best CSLR and SLT results occur at $\beta = 0.5$, suggesting equal emphasis on semantic matching and language modelling leads to balanced representations. For ISLR, optimal performance is achieved at $\beta = 0.6$, with a slight advantage from generative learning to capture discriminative features necessary for isolated recognition. Interestingly, the second-best ISLR performance is achieved at $\beta = 0.4$, where the model places greater weight on sign-text matching. It yields the same per-instance accuracy and slightly lower per-class accuracy, showing both objectives contribute meaningfully. These findings underscore the complementary nature of SGT matching and language modelling. Achieving an appropriate balance between the two is essential for optimising performance across a range of SLU tasks.

Table 4: Trade-off analysis between text matching and language modelling.

| Beta | WLASL2000 | | CSL-Daily | | | |
|---|---|---|---|---|---|---|
| | ISLR | | CSLR | | SLT | |
| | P-I↑ | P-C↑ | WER↓ | B@1↑ | B@4↑ | R@L↑ |
| 0.2 | 64.30 | 62.21 | 26.62 | 56.31 | 27.82 | 57.88 |
| 0.4 | 64.40 | 62.15 | 26.43 | 56.38 | 27.88 | 57.76 |
| **0.5** | 64.27 | 62.03 | **26.12** | **56.83** | **28.24** | **58.04** |
| **0.6** | **64.40** | **62.32** | 26.33 | 56.00 | 27.79 | 57.43 |
| 0.8 | 64.12 | 62.22 | 26.27 | 55.58 | 27.72 | 57.36 |

Table 5: ISLR results on WLASL2000 dataset.

| Methods | P-I↑ | P-C↑ |
|---|---|---|
| ST-GCN (Yan et al., 2018) | 34.40 | 32.53 |
| HMA (Hu et al., 2021b) | 37.91 | 35.90 |
| SignBERT (Zhou et al., 2021c) | 39.40 | 36.74 |
| BEST (Zhao et al., 2023) | 46.25 | 43.52 |
| SignBERT+ (Hu et al., 2023a) | 48.85 | 46.37 |
| MSLU (Zhou et al., 2024) | 56.29 | 53.29 |
| NLA-SLR (Zuo et al., 2023) | 61.05 | 58.05 |
| Uni-Sign (Li et al., 2025) | 63.52 | 61.32 |
| **Sigma** | **64.40** | **62.32** |

Additional ablation studies can be find in Appendix I.

## 4.4 CONTRIBUTION OF THE CORE COMPONENTS

Table 6 presents an ablation study evaluating the contribution of core components that inherited from pre-training. When both the sign encoder and SGT decoder are trained from scratch, the model performs poorly across all tasks, highlighting the inherent difficulty of SLU without structural guidance. Introducing individual components shows consistent improvement. Adding only the sign encoder largely reduces the CSLR WER from 523.85 to 180.85 and increases ISLR accuracy, indicating the ability of the sign encoder to model temporal visual patterns. Enabling both further boosts SLT performance, improving BLEU4 from 2.56 to 25.47 and ROUGE-L from 15.77 to 54.88, demonstrating the benefits of transferable visual representations learned in pre-training. Finally, inheriting parameters from all components yields the best performance across ISLR, CSLR, and SLT tasks, proving that the full architecture, integrating pre-training, a dedicated sign encoder, and a task-aware decoder, forms an effective pipeline for unified SLU.

Table 6: Impact of pre-training. A green check means the presence of pre-training or that parameters are inherited for fine-tuning; a red cross indicates no pre-training or that parameters are not inherited from pre-training.

| Pre-train | Sign encoder | SGT decoder | WLASL2000 ISLR | | CSL-Daily CSLR | SLT | | |
|---|---|---|---|---|---|---|---|---|
| | | | P-I↑ | P-C↑ | WER↓ | B@1↑ | B@4↑ | R@L↑ |
| ✗ | ✗ | ✗ | 0.02 | 0.01 | 523.85 | 7.14 | 0.12 | 11.95 |
| ✓ | ✗ | ✓ | 1.04 | 1.03 | 180.85 | 15.55 | 2.56 | 15.77 |
| ✓ | ✓ | ✗ | 22.56 | 20.41 | 27.83 | 52.79 | 25.47 | 54.88 |
| ✓ | ✓ | ✓ | **64.40** | **62.32** | **26.12** | **56.83** | **28.24** | **58.04** |

Table 7: CSLR results on CSL-Daily dataset.

| Methods | DEV WER↓ | TEST WER↓ |
|---|---|---|
| SignBT (Zhou et al., 2021a) | 33.20 | 33.20 |
| AdaBrowse (Hu et al., 2023e) | 31.20 | 30.70 |
| SEN (Hu et al., 2023d) | 31.10 | 30.70 |
| CorrNet (Hu et al., 2023c) | 30.60 | 30.10 |
| MSLU (Zhou et al., 2024) | 28.60 | 27.90 |
| CoSign (Jiao et al., 2023) | 28.10 | 27.20 |
| Uni-Sign (Li et al., 2025) | 26.70 | 26.00 |
| **Sigma** | **26.12** | **25.92** |

## 5 COMPARISON WITH STATE-OF-THE-ART METHODS

We evaluate Sigma across the three aforementioned core SLU tasks. For ISLR on the WLASL2000 dataset, our model sets a new performance benchmark (see Table 5). These results demonstrate strong gesture recognition and effective feature discrimination. For CSLR on the CSL-Daily dataset, as shown in Table 7, Our method achieves new state-of-the-art (SOTA) performance, surpassing the strong pose-RGB-based Uni-Sign model, highlighting improved temporal modelling and more precise alignment between sign sequences and sign glosses relying solely on skeletal data. For SLT (see Table 9 and Table 8), Sigma shows strong performance across How2Sign, OpenASL, and CSL-Daily. On How2Sign, it delivers improvements across all evaluation metrics. Sigma achieves new SOTA results on OpenASL across all evaluation metrics used for this study. On CSL-Daily, it surpasses all gloss-free methods and rivals the long-standing gloss-based SOTA model CV-SLT. These results confirm the generalisability of Sigma across varied datasets and SLU tasks. The complete results are in Appendix J.

Table 8: SLT Results on How2Sign and OpenASL.

| Methods | TEST B@1↑ | B@4↑ | R@L↑ |
|---|---|---|---|
| How2Sign | | | |
| GloFE-VN (Lin et al., 2023) | 14.90 | 2.20 | 12.60 |
| YouTube-ASL (Uthus et al., 2023) | 37.80 | 12.40 | - |
| MSLU (Zhou et al., 2024) | 20.10 | 2.40 | 17.20 |
| SLT-IV (Tarrés et al., 2023) | 34.00 | 8.00 | - |
| $C^2RL$ (Chen et al., 2025) | 29.10 | 9.40 | 27.00 |
| FLa-LLM (Chen et al., 2024) | 29.80 | 9.70 | 27.80 |
| **Sigma** | **40.06** | **15.61** | **36.71** |
| OpenASL | | | |
| GloFE-VN (Lin et al., 2023) | 21.56 | 7.06 | 21.75 |
| Conv-GRU (Camgoz et al., 2018) | 16.11 | 4.58 | 16.10 |
| I3D-transformer (Shi et al., 2022) | 18.31 | 5.66 | 18.64 |
| OpenASL (Shi et al., 2022) | 20.92 | 8.59 | 21.02 |
| Uni-Sign (Li et al., 2025) | 49.35 | 23.14 | 43.22 |
| $C^2RL$ (Chen et al., 2025) | 31.46 | 13.21 | 31.36 |
| **Sigma** | **49.55** | **23.19** | **44.47** |

Table 9: SLT results on CSL-Daily dataset.

| Methods | DEV B@1↑ | B@4↑ | R@L↑ | TEST B@1↑ | B@4↑ | R@L↑ |
|---|---|---|---|---|---|---|
| Gloss-based | | | | | | |
| SLRT (Camgoz et al., 2020) | 37.47 | 11.88 | 37.96 | 37.38 | 11.79 | 36.74 |
| ConSLT (Fu et al., 2023) | - | 14.80 | 41.46 | - | 14.53 | 40.98 |
| SignBT (Zhou et al., 2021a) | 51.46 | 20.80 | 49.49 | 51.42 | 21.34 | 49.31 |
| MMTLB (Chen et al., 2022a) | 53.81 | 24.42 | 53.38 | 53.31 | 23.92 | 53.25 |
| SLTUNET (Zhang et al., 2023) | - | 23.99 | 53.58 | 54.98 | 25.01 | 54.08 |
| TS-SLT (Chen et al., 2022b) | 55.21 | 25.76 | 55.10 | 55.44 | 25.79 | 55.72 |
| CV-SLT (Zhao et al., 2024a) | 56.36 | **28.24** | 56.36 | 58.29 | **28.94** | 57.06 |
| Gloss-free | | | | | | |
| SLRT (Camgoz et al., 2020) | 21.03 | 4.04 | 20.51 | 20.00 | 3.03 | 19.67 |
| GASLT (Yin et al., 2023) | - | - | - | 19.90 | 4.07 | 20.35 |
| MSLU (Zhou et al., 2024) | 33.28 | 10.27 | 33.13 | 33.97 | 11.42 | 33.80 |
| NSLT (Camgoz et al., 2018) | 34.22 | 7.96 | 34.28 | 34.16 | 7.56 | 34.54 |
| GFSLT-VLP (Zhou et al., 2023) | 39.20 | 11.07 | 36.70 | 39.37 | 11.00 | 36.44 |
| FLa-LLM (Chen et al., 2024) | - | - | - | 37.13 | 14.20 | 37.25 |
| $C^2RL$ (Chen et al., 2025) | - | - | - | 49.32 | 21.61 | 48.21 |
| Uni-Sign (Li et al., 2025) | 55.30 | 26.25 | 56.03 | 55.08 | 26.36 | 56.51 |
| SignLLM (Gong et al., 2024) | 42.45 | 12.23 | 39.18 | 39.55 | 15.75 | 39.91 |
| Sign2GPT (Wong et al., 2024) | - | - | - | 41.75 | 15.40 | 42.36 |
| **Sigma** | **56.83** | **28.24** | **58.04** | **55.97** | **27.30** | **57.58** |

## 6 CONCLUSION

SLU requires the ability to recognise fine-grained visual patterns while simultaneously modelling complex linguistic semantics. We identify three key challenges in current SLU: 1) weak semantic grounding in visual features, 2) imbalanced local-global modelling, and 3) inadequate cross-modal alignment. To address these, we propose Sigma, a skeleton-based unified framework for semantically informative and transferable representation learning for SLU. Sigma introduces: 1) the SignEF mechanism for bidirectional visual-textual interaction during encoding, 2) the SignEF strategy for optimising local and global alignment via contrastive objectives, and 3) a unified pre-training scheme combining contrastive learning, text matching, and language modelling. We validate the effectiveness of Sigma on multiple SLU benchmarks, including WLASL, CSL-Daily, How2Sign, and OpenASL. Sigma consistently exhibits strong performance across the aforementioned SLU tasks. These results underscore the importance of semantically informed pre-training for building scalable and robust skeleton-based SLU model.

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

CONTENTS

# A    MOTIVATION

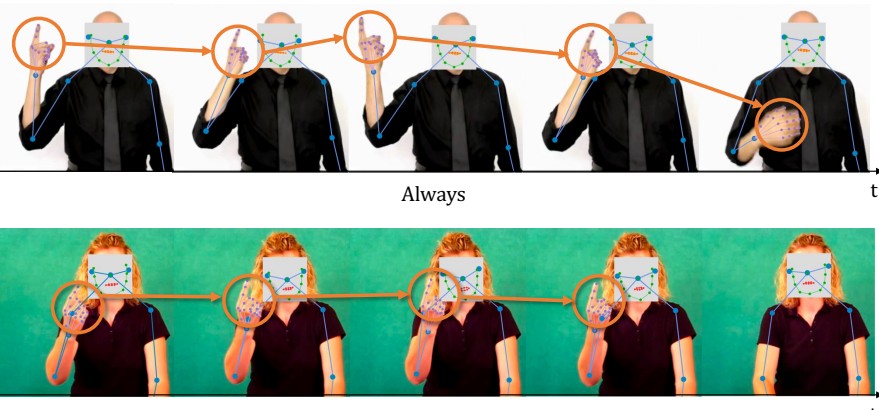

Figure 5: Visualization derived from the WLASL2000 dataset. The right hand, along with its primary motion trajectory, is highlighted to illustrate the gesture dynamics. The figure shows two sign sequences, "Always" and "Someone." Although both gestures exhibit similar hand shapes and motion trajectories, they differ in spatial and temporal extent. Disambiguating them requires not only local visual detail but also global temporal understanding and accurate alignment with linguistic meaning, highlighting the need for effective multimodal representation learning.

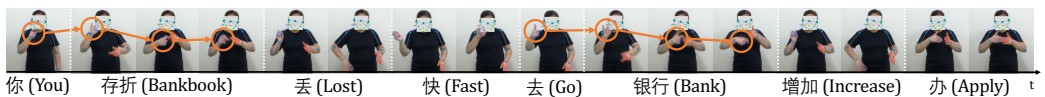

Figure 6: Visualization derived from the CSL-Daily dataset. The right hand, along with its primary motion trajectory, is highlighted to illustrate the gesture dynamics. The example corresponds to the sentence: "存折丢了的话要马上去银行补办。" (If your bankbook is lost, you should go to the bank immediately to have it reissued.) This figure illustrates visually similar signs such as "bankbook" and "bank", as well as "you" and "go". Despite sharing highly similar motion patterns, each gesture serves a distinct syntactic and semantic function within the sentence. This example demonstrates the limitations of purely visual recognition and emphasizes the importance of strong visual-linguistic alignment for effective SLU.

We identified three key challenges faced by current SLP-based SLU methods in Section 1: weak semantic grounding, imbalanced local-global feature modelling, and ineffective cross-modal alignment. These issues frequently manifest in practical scenarios where visually similar gestures convey entirely different meanings depending on their context, temporal structure, or semantic function. Figures 5 and 6 provide visual examples drawn from the WLASL2000 and CSL-Daily datasets, respectively, illustrating how these challenges affect SLU. Note that the figures show selected frames for clarity, rather than the full sequence.

## A.1    WEAK SEMANTIC GROUNDING.

In the CSL-Daily example shown in Figure 6, the sign sequence includes terms such as "bankbook" and "bank", as well as "you" and "go", which share similar hand shapes and spatial trajectories. Although these gestures appear visually alike, each one conveys a distinct meaning and serves a different syntactic function within the sentence. If a model focuses only on superficial motion or shape patterns without understanding the linguistic intent behind each gesture, it may generate inaccurate or overly generic translations. This example emphasizes the importance of semantic grounding, where models should recognize what is being signed and understand its meaning within the broader linguistic and contextual framework.

## A.2 LOCAL-GLOBAL IMBALANCE.

The WLASL2000 examples shown in Figure 5 present two sign sequences, "Always" and "Someone," which share highly similar hand shapes and motion trajectories across several frames. The primary distinction lies in the broader spatial and temporal extent of "Always" compared to the more confined gesture of "Someone." Relying solely on local visual cues such as hand configuration or position is insufficient for accurate interpretation. At the same time, global cues alone cannot resolve subtle variations in form that are crucial for meaning. Accurate understanding requires the integration of fine-grained local details with the overarching motion pattern and semantic context. This example underscores the essential role of modelling both local and global features together. Only by combining detailed gesture recognition with a coherent understanding of the full temporal sequence can models distinguish between signs that are visually similar but semantically different.

## A.3 INEFFECTIVE CROSS-MODAL ALIGNMENT.

Although Figures 5 and 6 highlight different challenges, both reveal a deeper problem rooted in weak alignment between visual and textual modalities. In Figure 5, distinguishing between "Always" and "Someone" involves more than recognizing visual patterns. It requires establishing a clear connection between the motion sequence and its corresponding linguistic meaning. Similarly, in Figure 6, the model should determine whether a gesture refers to "bank" or "bankbook," even when the visual cues appear highly similar. Accurate interpretation depends on correctly linking each visual segment to its intended word or phrase within a broader sentence. Without a strong mechanism for aligning gestures with language, the model fails to generate consistent and meaningful outputs. These examples show that SLU is not just a visual recognition problem; it is a multimodal challenge that requires precise mapping from gestures to language at both lexical and semantic levels.

These visualizations serve as motivating evidence for the limitations (as discussed in Section 1) of existing SLU approaches and the need for semantically informed modelling. Our proposed framework mitigates the impact of these problems by enriching visual features with linguistic context, balancing local and global feature interactions, and learning aligned cross-modal representations.

## B ADVANCES IN SKELETON-BASED SLU

A growing line of work explores skeleton as a compact and semantically informative modality for SLU. Early work such as GCN-BERT (Tunga et al., 2021) integrates graph convolutional networks over human joint graphs with transformer encoders, modelling spatial–temporal cues from skeletal sequences. Although effective for isolated SLR, it remains fully supervised and task-specific. To the best of our acknowledged, SignBERT Hu et al. (2021a) is the first work applies self-supervised pre-training for hand-centric skeletal representations. By masking and reconstructing hand trajectories and leveraging an explicit hand-shape model for regularisation, it learns richer visual embeddings and improves both isolated and continuous recognition. Building on stronger structural modelling, BEST (Zhao et al., 2023) advances skeleton-based pre-training by grouping body and hands into skeletal triplets and adopting a BERT-style masked unit modelling approach. A discrete VAE is used to tokenise continuous skeletal units into pseudo-tokens, enabling cross-entropy reconstruction and encouraging contextual reasoning over articulated hand–body interactions. BEST (Zhao et al., 2023) demonstrates strong generalisation across isolated SLR benchmarks. SignBERT+ Hu et al. (2023b) further extends this family of work by incorporating linguistic signals, and refining the pre-training tasks to support SLR and SLT. Compared with its predecessor, it provides more structured multi-task learning and better alignment between skeletal sequences and semantics. These methods reveal the strong potential of skeleton-only pre-training. However, they focus on capturing visual cues, lack joint modeling on the textual information, and remain visually grounded and largely task-specific. In contrast, Sigma builds on this foundation while moving toward a unified SLU paradigm and learns cross-modal alignment suitable for ISLR, CSLR, and SLT within a single framework.

## C  QUALITATIVE ANALYSIS

To further validate the semantic advantages of our proposed method, we present qualitative results derived for all benchmark datasets used in this study. Each table contrasts ground-truth references with results outputted by our method (Sigma).

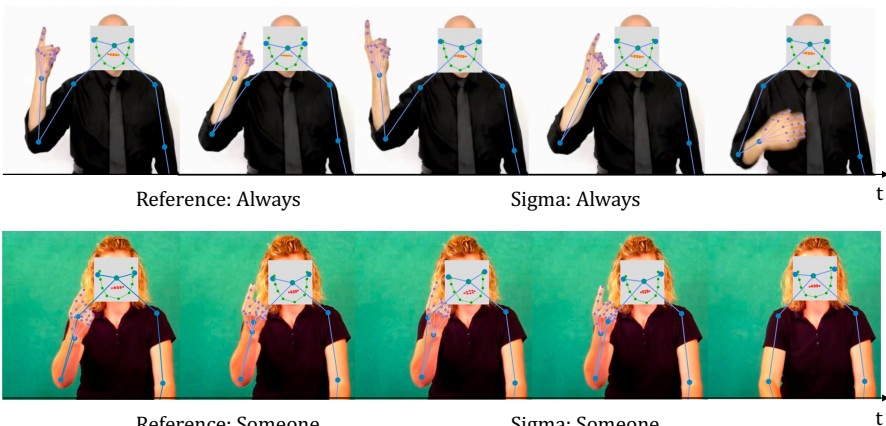

Figure 7: Qualitative examples derived from the WLASL2000 dataset for ISLR.

For ISLR, Figure 7 presents qualitative examples from the WLASL2000 dataset. Sigma demonstrates consistent recognition across signers with varying visual appearances and signing styles. This stability reflects the improved semantic grounding and more effective cross-modal alignment, which together help mitigate influence caused by subtle differences in gesture execution. By capturing both fine-grained motion details and broader temporal structure, Sigma supports more reliable recognition, aligning with our objective of balancing local precision with global contextual understanding.

In the examples of each table, our method achieves complete correctness, replicating references. Although these illustrate baseline competence, the deeper value of our framework emerges in the challenging examples, those where slight variations occur between the predictions of the model and the ground-truth. Instead of treating these discrepancies as outright errors, we analyze them through the lens of semantic grounding and linguistic alignment.

Table 10: Qualitative examples derived from the CSL-Daly dataset for CSLR.

| | |
|---|---|
| Reference: | 椅子 他们 想 什么 时间 去 买 |
| Sigma: | 椅子 他们 想 什么 时间 去 买 |
| Reference: | 帮助 看 这 衣服 怎么样 |
| Sigma: | 帮助 看 这 衣服 怎么样 |
| Reference: | 可以 这 近 不 远 饭店 走 多少 到 |
| Sigma: | 可以 这 近 不 远 饭店 走 多少 到 |
| Reference: | 计算 结果 我们 必须 要 准确 |
| Sigma: | 计算 结果 我们 必须 准确 |
| Reference: | 存折 丢 快 去 银行 增加 办 |
| Sigma: | 你 存折 丢 快 去 银行 增加 办理 |
| Reference: | 核磁共振 方法 来 决定 表面 机器 |
| Sigma: | 核磁共振 方法 来 固定 封面 机器 |

For CSLR, as shown in the bottom rows of Table 10, Sigma exhibits semantic preservation despite minor lexical variations. For example, phrases like "决定 表面" and "固定 封面" differ in wording but convey similar meanings. While such variations reduce scores like WER, they are easily under­stood by human readers, as language naturally allows multiple ways to express the same idea. These

Table 11: Qualitative examples derived from the How2Sign dataset for SLT.

| | |
|---|---|
| Reference: | My name is Dr. Art Bowler. |
| Sigma: | My name is Dr. Art Bowler. |
| Reference: | What do you see? |
| Sigma: | What do you see? |
| Reference: | My name is Allen Diwan. |
| Sigma: | Hi, I'm Allen Diwan. |
| Reference: | You're having a good time along the way. |
| Sigma: | It's a really enjoyable process. |
| Reference: | Stay safe, and we'll see ya' next time. |
| Sigma: | See you next time. |
| Reference: | I hope you're having fun. |
| Sigma: | I hope you had fun with it. |

cases highlight the enhanced semantic grounding: even when the predicted glosses deviate from the reference, the intended meaning remains intact. This is especially crucial in CSLR, where explicit gloss segmentation is absent and contextual understanding plays a key role.

Table 12: Qualitative examples derived from the OpenASL dataset for SLT.

| | |
|---|---|
| Reference: | America! |
| Sigma: | America! |
| Reference: | I'm from Austin, Texas! |
| Sigma: | I'm from Austin, Texas! |
| Reference: | See you at the conference this July! |
| Sigma: | See you at the conference this July! |
| Reference: | That's not right! |
| Sigma: | That's not fair. |
| Reference: | Also, be sure you talk to your legislators. |
| Sigma: | You will also talk with your legislators. |
| Reference: | Progress is being made. |
| Sigma: | The work is moving forward. |

For SLT, qualitative examples from English-based datasets (Tables12 and 11) demonstrate the ability of the model to produce fluent and contextually appropriate sentences and the generalization of the model across speaker identities as well as conversational styles. For instance in Table 11, it converts "I hope you're having fun." into "I hope you had fun with it." showing an understanding of tense and implied context. These changes demonstrate that the model captures deeper semantic meaning rather than relying only on surface-level similarity. In Table 12, the model outputs "You will also talk with your legislators" instead of the reference "Also, be sure you talk to your legislators." Though not a verbatim match, the generated sentence is syntactically sound and preserves the core message, demonstrating sentence-level comprehension. This reflects the impact of our SignEF strategy in bridging local-global semantics across modalities. In Table 13, the Chinese examples show similar results. In the fourth row, "他醒来时发现自己在医院里" and "他醒来后，发现自己在医院" use different syntax but express the same idea. In the fifth example, "需要回家拿" and "回家要拿" describe the same action with a slight variation in sentence structure. In the final row, the model adds a rhetorical question "你想喝什么？", which enhances the sentence without changing its meaning. These variations may affect token-based metrics, but they reflect natural language use and communicative clarity.

Table 13: Qualitative examples derived from the CSL-Daily dataset for SLT.

| | |
|---|---|
| Reference: | 我 每 天 六 点 起 床 。 |
| Sigma: | 我 每 天 六 点 起 床 。 |
| Reference: | 警 察 要 检 查 你 的 身 份 证 。 |
| Sigma: | 警 察 要 检 查 你 的 身 份 证 。 |
| Reference: | 苹 果 是 你 买 的 吗 ？ |
| Sigma: | 苹 果 是 你 买 的 吗 ？ |
| Reference: | 他 醒 来 时 发 现 自 己 在 医 院 里 。 |
| Sigma: | 他 醒 来 后 , 发 现 自 己 在 医 院 。 |
| Reference: | 我 的 护 照 忘 记 带 了 , 需 要 回 家 拿 。 |
| Sigma: | 我 的 护 照 忘 带 了 , 回 家 要 拿 。 |
| Reference: | 桌 上 放 着 很 多 饮 料 , 你 喝 什 么 ？ |
| Sigma: | 桌 子 上 有 很 多 饮 料 , 你 想 喝 什 么 ？ |

## D  CLUSTER AGGREGATOR

To address the challenges of weak semantic grounding and limited cross-modal alignment, we design a cluster aggregator module inspired by (Hou et al., 2024) to produce cluster-level textual embeddings that better correspond to visual sign units. Given a text input such as "curiosity," the tokenizer splits it into subword tokens, which are then processed by the text encoder to generate token-level features. The aggregator groups these features into semantically coherent clusters. An offset calculator maps each original token to its cluster index, and the aggregation helper combines features within each cluster to form a compact representation. This process yields text embeddings that preserve semantic structure while reducing redundancy. Sigma supports both fine-grained gesture recognition and high-level translation. This mechanism contributes to more effective cross-modal learning and helps bridge the gap between dynamic visual input and structured language representations.

## E  SIGN-GROUNDED TEXT ENCODER

To mitigate the impact of weak semantic grounding and ineffective cross-modal alignment, we enhance text representations by integrating visual cues from sign features through a dual-path architecture. This SGT encoder consists of two parallel branches: a sign-text matching (STM) path and a language modelling (LM) path. The STM path, repeated $M$ times, cooperates with cross-attention layers where textual tokens attend to sign features, allowing the model to align linguistic units with visual semantics and enrich textual embeddings with SL gesture context. The LM path, repeated $N$ times, uses standard transformer blocks with self-attention and feed-forward layers to preserve language fluency and syntactic structure. This dual-path setup enables the SGT encoder to learn representations that are both semantically grounded in visual input and linguistically coherent. During fine-tuning, all parameters except for the self-attention layers within the**STM** path are transferred, ensuring effective knowledge reuse while allowing flexible adaptation to downstream SLU tasks. This design supports stronger cross-modal alignment and helps mitigate the semantic disconnect between dynamic sign inputs and static textual outputs.

## F  SKELETAL DATA

The sign sequences are skeletal data extracted using RTMPose (Jiang et al., 2023) from MMPose (Sengupta et al., 2020). Figure 8 illustrates the visualization of 69 keypoints per frame, including 21 for each hand, 9 for the body, and 18 for the face.

The table 14 summarises four benchmark sign language datasets in terms of language, language level, number of samples, and storage size for both RGB and skeletal data. WLASL, How2Sign, and OpenASL are American sign language, and CSL Daily is Chinese sign language. Together they

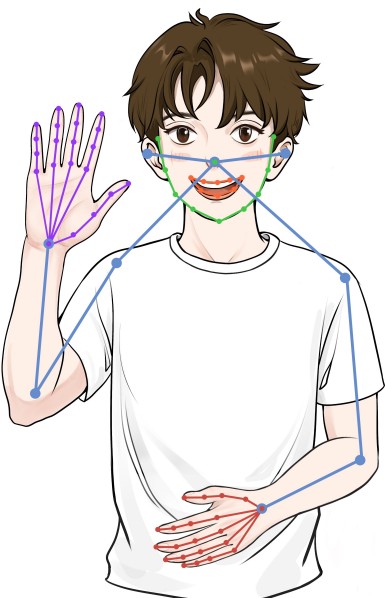

Figure 8: The visualisation of the full-body keypoints.

span both gloss and sentence level, with sample counts ranging from 20,654 to 98,419. While the RGB videos are large in storage size, the skeletal data is far more compact, with sizes reduced by an order of magnitude. This compactness translates into faster loading times and lower computational overhead, making skeletal data more scalable and efficient for model training or deployment with user preference in gestures. Beyond efficiency, skeletal representations also abstract away background noise and highlight body motion dynamics, thereby preserving linguistic cues that are critical for SLU. Thus, the table not only illustrates dataset diversity in scale and annotation level but also underscores the practical advantages of skeletal data for efficient and robust SLU.

Table 14: Statistics of benchmark datasets

| Dataset | Language | Level | # Samples | Size (RGB, GB) | Size (Skeleton, GB) |
|---------|----------|-------|-----------|----------------|---------------------|
| WLASL | American | Gloss | 21,083 | 78.84 | 3.68 |
| How2Sign | American | Sentence | 35,263 | 329.00 | 15.58 |
| OpenASL | American | Sentence | 98,419 | 638.03 | 29.78 |
| CSL-Daily | Chinese | Sentence | 20,654 | 92.80 | 4.27 |

This table 15 compares RGB and skeletal modalities in terms of average file size per sample, and time required to load a single sample of both modalities. Skeletal data significantly reduces storage and loading time, making it more suitable for efficient training and real-world deployment.

Table 15: Comparison of RGB and skeletal data.

| Modality | Avg. size per sample | Loading time per sample |
|----------|---------------------|-------------------------|
| RGB | 4714.84 KB | 455.35 ms |
| Skeletal | 437.18 KB | 9.58 ms |

## G WHAT MAKES ANNOTATIONS COSTLY IN SIGN LANGUAGE PROCESSING?

In sign language processing, annotations refer to manually labelled data that describe the content and structure of SL videos. These annotations are essential for training supervised learning models, but are significantly more expensive and labour-intensive than those in natural language processing.

There are three main reasons why annotations in this domain are costly:

1) **Expert-dependent labelling :** Unlike speech or text, SL does not have a widely standardised written form. Annotators must label each gesture with its corresponding gloss, a textual representation of the meaning of the sign. This requires a deep level of linguistic expertise in both the SL and the spoken language to which it is assigned. It is time-consuming, and the availability of such annotations is limited.

2) **Temporal segmentation and alignment:** For CSLR and SLT tasks, annotators must align glosses with precise time frames in SL videos. Unlike tokenising text, this process requires identifying the exact start and end points of each sign within a continuous, unsegmented motion stream. Such fine-grained temporal labelling demands both visual precision and linguistic expertise, making the task exceptionally labour-intensive. In our study, temporal boundary labels are not used; glosses are only employed for ISLR and CSLR. With the growing availability of public SL datasets, we hope that both ISLR and CSLR can eventually be learned without relying on any costly gloss annotations.

3) **Multi-layer multimodal cues:** SL relies on hand gestures, facial expressions, body posture, and spatial references. Annotating these multimodal components accurately requires frame-by-frame observation and sometimes multi-camera viewpoints. Capturing this richness adds both time and complexity to the annotation process.

Due to these factors, building large-scale annotated datasets for SLU or SL tasks remains a major bottleneck. This motivates the development of SLP-based SLU models as well as the use of self-supervised and weakly supervised methods, which can learn meaningful representations from unannotated or minimally annotated data.

# H RETHINKING THE ROLE OF GLOSSES IN SLU

Gloss annotations have long been used as an intermediate representation in sign language translation, and they provide efficient and powerful supervision. By reducing the gap between raw visual input and spoken language output, glosses offer a structured, linguistically meaningful signal that has boosted SLT performance compared to purely end-to-end gloss-free approaches (Zhou et al., 2023). At the same time, this benefit comes with significant drawbacks that increasingly limit scalability and linguistic fidelity. First, glosses are costly to obtain and difficult to scale. Producing them requires expert annotators and fine-grained temporal alignment, making data collection expensive and slow, and constraining the size of available datasets. Second, glosses act as an information bottleneck. A gloss sequence compresses rich and continuous sign expressions into discrete tokens, discarding nuances not represented in the gloss inventory and weakening the direct mapping from visual input to textual semantics. Third, gloss-based pipelines suffer from error propagation and mismatched objectives. Since they typically rely on a continuous sign language recognition front end, recognition errors are passed into the translation stage, while training remains split across recognition and translation tasks rather than being optimised jointly. Moreover, glosses often fail to capture divergences between sign language structure and the linear order of spoken languages. By committing to a single gloss sequence early, models risk locking in alignment hypotheses that hinder later reordering and discourse modelling. Another limitation lies in the inability of glosses to encode non-manual signals such as facial expressions, or the compositional use of multiple articulators, both of which carry critical linguistic meaning. In addition, gloss conventions vary across datasets and languages, introducing inconsistencies that reflect annotation practices rather than genuine linguistic differences, which in turn hinders transfer learning and cross-corpus pre-training. Finally, reliance on gloss labels restricts data efficiency. Although gloss supervision can enhance performance when available, it blocks the use of large unlabelled video corpora, whereas direct visual–text modelling allows learning from broader resources. Taken together, these issues highlight that glosses provide strong but narrow supervision: while effective in guiding alignment, they remain a bottleneck for scalability. This motivates our use of cluster-wise contrastive learning, which produces gloss-like groupings automatically and retains the advantages of structured alignment while avoiding the limitations of manual gloss annotation.

# I ADDITIONAL EXPERIMENT

## I.1 OPTIMISING LOCAL CLUSTER-WISE CONTRASTIVE LEARNING STRATEGIES

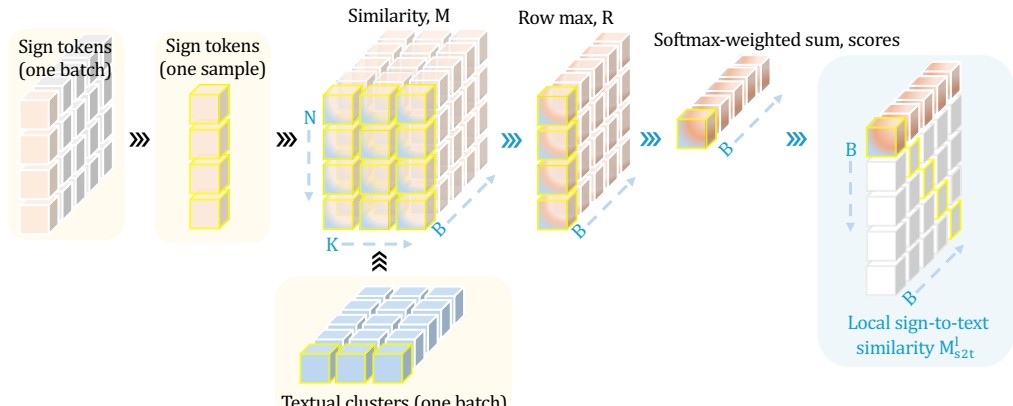

Figure 9: Illustration of the computation of our local sign-to-text cluster-wise similarity inspired by (Chen et al., 2020; Radford et al., 2021; Li et al., 2022a; Hou et al., 2024). The similarity matrix $M$ is computed between the sign tokens of each sample and all textual clusters. For each row, the maximum similarity value is obtained using a row max operation. The resulting values are passed through a softmax-weighted sum function to obtain the local similarity scores. Finally, in-batch local cluster contrastive learning is applied to pull semantically aligned visual-text pairs (highlighted by light yellow borders) closer, while pushing apart unaligned pairs. This process enables localised semantic grounding by focusing on the most relevant visual-text associations within each cluster.

### I.1.1 ROW-WISE OPERATIONS

To compute cluster-wise similarity (as illustrated in Figure 9) in an optimal way, we evaluate several row-wise operations in Table 16, including row max, average, top-$k$ average, and softmax-based operation. For the top $k$ average, the value of $k$ is dynamically determined based on the number of clusters or tokens that exist in the cosine similarity matrix $M$, using the formula:

$$k = \max\left(1, \lfloor \tfrac{M}{3} \rfloor\right)$$

This ensures that $k$ remains a valid positive integer bounded by the length of the last dimension of $M$, with a lower bound of 1 to avoid degenerate cases. In addition to max and average operation over the similarity matrix $M$, we also evaluated a softmax-based operation expressed as:

$$R = \text{sum}(\text{softmax}(M) \odot M, \ dim = 1)$$

Empirically, our ablation results (as listed in the Table 16) show that row max consistently yields optimal performance across tasks. For instance, on CSLR (CSL Daily), the max operation achieves a WER of 26.12, compared to 26.70 with row average, 26.12 with top k average, and 26.59 with softmax. Similarly, for SLT (CSL Daily), max results in a BLEU 4 score of 28.24, outperforming 27.28 (average), 27.84 (top k average), and 27.12 (softmax).

These findings suggest that while top-$k$ average and softmax aim to balance or smooth cluster-level similarities, they tend to dilute the most salient alignment cues. In contrast, the max operation emphasises the strongest cluster level signals, providing sharper contrastive gradients and thus more discriminative cross-modal alignment. This justifies our decision to adopt row wise max as the default in our local contrastive learning.

### I.1.2 LOCAL-LEVEL SCORING

To score similarity across visual tokens or textual clusters, we investigate several scoring methods in Table 17. While simple summation and averaging provide basic baselines, they often fail to adequately emphasise the most informative local level correspondences. Therefore, we highlight three more

Table 16: Row operations for local cluster-wise contrastive learning.

| Strategy | WLASL2000 | | CSL-Daily | | | |
|---|---|---|---|---|---|---|
| | ISLR | | CSLR | | SLT | |
| | P-I↑ | P-C↑ | WER↓ | B@1↑ | B@4↑ | R@L↑ |
| **Row max** | **64.40** | **62.32** | **26.12** | **56.83** | **28.24** | **58.04** |
| Row average | 64.20 | 61.91 | 26.70 | 55.95 | 27.28 | 56.81 |
| Row top-$k$ average | 64.32 | 62.32 | 26.12 | 56.36 | 27.84 | 57.52 |
| Row softmax | 64.27 | 62.06 | 26.59 | 56.24 | 27.12 | 56.66 |

expressive alternatives: softmax, log-sum-exp and variance-reduced-sum, which offer improved semantic sensitivity presented below. This softmax method serves as our primary scoring method and consistently outperforms the others by dynamically weighting token pairs. The probabilistic weighting of token similarity provided by softmax enables the model to emphasise informative alignments while still preserving contextual diversity. Although log-sum-exp and variance-reduced-sum perform competitively, they exhibit larger variability across benchmarks. Therefore, we adopt softmax as our default accumulation method, as it offers a reliable and generalisable approach for semantic alignment in cluster-wise contrastive learning.

Table 17: Local-level scoring methods.

| Scoring Method | Pseudocode |
|---|---|
| Softmax | $score \leftarrow \text{sum}(\text{softmax}(R) \odot R,\ dim = 1)$ |
| Log-sum-exp | $score \leftarrow \log(\text{sum}(\exp(R),\ dim = 1))$ |
| Variance-reduced-sum | $score \leftarrow \text{sum}((R - \text{mean}(R,\ dim = 1)),\ dim = 1)$ |

Table 18: Local-level scoring methods for the local cluster-wise contrastive learning.

| Strategy | WLASL2000 | | CSL-Daily | | | |
|---|---|---|---|---|---|---|
| | ISLR | | CSLR | | SLT | |
| | P-I↑ | P-C↑ | WER↓ | B@1↑ | B@4↑ | R@L↑ |
| Sum | 64.22 | 62.00 | 26.64 | 55.46 | 28.10 | 57.43 |
| Average | 64.17 | 62.06 | 27.06 | 56.24 | 27.67 | 57.15 |
| Log-sum-exp | 64.37 | 62.28 | 26.12 | 56.41 | 27.97 | 58.08 |
| **Softmax** | **64.40** | **62.32** | **26.12** | **56.83** | **28.24** | **58.04** |
| Variance-reduced-sum | 64.35 | 62.08 | 26.38 | 56.25 | 27.47 | 57.03 |

## I.2 SHOULD THE TEXT ENCODER BE TRAINABLE IN SIGMA'S PRE-TRAINING?

To evaluate the role of the text encoder during pre-training, we compare the effects of freezing and unfreezing its parameters, as shown in Table 19. While the improvements are relatively small, unfreezing the text encoder consistently leads to better performance across all SLU tasks. On CSL-Daily, it lowers the CSLR WER and yields marginal gains in SLT metrics such as BLEU1, BLEU4, and ROUGE-L. ISLR also shows improvements in both per-instance and per-class accuracy. These findings suggest that allowing the text encoder to update during pre-training would support better adaptation to visual features, contributing to more coherent cross-modal representations. This adjustment, though modest, would offer broader benefits beyond the specific benchmarks used in this study.

## I.3 COOPERATE DIFFERENT MODALITIES DURING PRE-TRAINING

For the sake of understanding how different input modalities contribute during pre-training, we investigate the effect of using either sign features or text features as inputs to the cross-attention module

Table 19: Impact of freezing text encoder during pre-training.

| Text encoder | WLASL2000 | | CSL-Daily | | | |
|---|---|---|---|---|---|---|
| | ISLR | | CSLR | SLT | | |
| | P-I↑ | P-C↑ | WER↓ | B@1↑ | B@4↑ | R@L↑ |
| Freezed | 64.32 | 62.19 | 26.49 | 56.62 | 27.63 | 57.03 |
| **Unfreezed** | **64.40** | **62.32** | **26.12** | **56.83** | **28.24** | **58.04** |

Table 20: Impact of different feature modalities.

| Features | WLASL2000 | | CSL-Daily | | | |
|---|---|---|---|---|---|---|
| | ISLR | | CSLR | SLT | | |
| | P-I↑ | P-C↑ | WER↓ | B@1↑ | B@4↑ | R@L↑ |
| **Sign feature** | **64.40** | **62.32** | **26.12** | **56.83** | **28.24** | 58.04 |
| Text feature | 64.37 | 62.15 | 26.12 | 56.82 | 28.21 | 58.12 |

Table 21: SLT results on OpenASL dataset.

| Method | DEV | | | | | TEST | | | | |
|---|---|---|---|---|---|---|---|---|---|---|
| | B@1 | B@2 | B@3 | B@4 | R@L | B@1 | B@2 | B@3 | B@4 | R@L |
| GloFE-VN (Lin et al., 2023) | 21.06 | 12.34 | 8.68 | 6.68 | 21.37 | 21.56 | 12.74 | 9.05 | 7.06 | 21.75 |
| Conv-GRU (Camgoz et al., 2018) | 16.72 | 8.95 | 6.31 | 4.82 | 16.25 | 16.11 | 8.85 | 6.18 | 4.58 | 16.10 |
| I3D-transformer (Shi et al., 2022) | 18.26 | 10.26 | 7.17 | 5.60 | 18.88 | 18.31 | 10.15 | 7.19 | 5.56 | 18.64 |
| OpenASL (Shi et al., 2022) | 20.10 | 11.81 | 8.43 | 6.57 | 20.43 | 20.92 | 12.08 | 8.59 | 6.72 | 21.02 |
| Uni-Sign (Li et al., 2025) | 50.84 | 37.82 | 29.83 | 24.16 | 44.58 | 49.35 | 36.32 | 28.55 | 23.14 | 43.22 |
| $C^2RL$ (Chen et al., 2025) | - | - | - | - | - | 31.46 | 21.85 | 16.58 | 13.21 | 31.36 |
| **Sigma** | **51.35** | **38.67** | **30.88** | **25.03** | **46.13** | **49.55** | **36.52** | **28.74** | **23.19** | **44.47** |

Table 22: SLT results on How2Sign dataset.

| Method | TEST | | | | |
|---|---|---|---|---|---|
| | B@1 | B@2 | B@3 | B@4 | R@L |
| GloFE-VN (Lin et al., 2023) | 14.90 | 7.30 | 3.90 | 2.20 | 12.60 |
| YouTube-ASL (Uthus et al., 2023) | 37.80 | 24.10 | 16.90 | 12.40 | - |
| MSLU (Zhou et al., 2024) | 20.10 | 7.70 | - | 2.40 | 17.20 |
| SLT-IV (Tarrés et al., 2023) | 34.00 | 19.30 | 12.20 | 8.00 | - |
| $C^2RL$ (Chen et al., 2025) | 29.10 | 18.60 | 12.90 | 9.40 | 27.00 |
| FLa-LLM (Chen et al., 2024) | 29.80 | 19.00 | 13.30 | 9.70 | 27.80 |
| **Sigma** | **40.06** | **27.48** | **20.30** | **15.61** | **36.71** |

Table 23: SLT results on CSL-Daily dataset.

| Method | DEV | | | | | TEST | | | | |
|---|---|---|---|---|---|---|---|---|---|---|
| | B@1 | B@2 | B@3 | B@4 | R@L | B@1 | B@2 | B@3 | B@4 | R@L |
| Gloss-based | | | | | | | | | | |
| SLRT (Camgoz et al., 2020) | 37.47 | 24.67 | 16.86 | 11.88 | 37.96 | 37.38 | 24.36 | 16.55 | 11.79 | 36.74 |
| ConSLT (Fu et al., 2023) | - | - | - | 14.80 | 41.46 | - | - | - | 14.53 | 40.98 |
| SignBT (Zhou et al., 2021a) | 51.46 | 37.23 | 27.51 | 20.80 | 49.49 | 51.42 | 37.26 | 27.76 | 21.34 | 49.31 |
| MMTLB (Chen et al., 2022a) | 53.81 | 40.84 | 31.29 | 24.42 | 53.38 | 53.31 | 40.41 | 30.87 | 23.92 | 53.25 |
| SLTUNET (Zhang et al., 2023) | - | - | - | 23.99 | 53.58 | 54.98 | 41.44 | 31.84 | 25.01 | 54.08 |
| TS-SLT (Chen et al., 2022b) | 55.21 | 42.31 | 32.71 | 25.76 | 55.10 | 55.44 | 42.59 | 32.87 | 25.79 | 55.72 |
| CV-SLT (Zhao et al., 2024a) | **58.05** | **44.73** | **35.14** | **28.24** | **56.36** | **58.29** | **45.15** | **35.77** | **28.94** | **57.06** |
| Gloss-free | | | | | | | | | | |
| SLRT (Camgoz et al., 2020) | 21.03 | 9.97 | 5.96 | 4.04 | 20.51 | 20.00 | 9.11 | 4.93 | 3.03 | 19.67 |
| GASLT (Yin et al., 2023) | - | - | - | - | - | 19.90 | 9.94 | 5.98 | 4.07 | 20.35 |
| MSLU (Zhou et al., 2024) | 33.28 | 21.31 | - | 10.27 | 33.13 | 33.97 | 22.20 | - | 11.42 | 33.8 |
| NSLT (Camgoz et al., 2018) | 34.22 | 19.72 | 12.24 | 7.96 | 34.28 | 34.16 | 19.57 | 11.84 | 7.56 | 34.54 |
| GFSLT-VLP (Zhou et al., 2023) | 39.20 | 25.02 | 16.35 | 11.07 | 36.70 | 39.37 | 24.93 | 16.26 | 11.00 | 36.44 |
| FLa-LLM (Chen et al., 2024) | - | - | - | - | - | 37.13 | 25.12 | 18.38 | 14.20 | 37.25 |
| $C^2RL$ (Chen et al., 2025) | - | - | - | - | - | 49.32 | 36.28 | 27.54 | 21.61 | 48.21 |
| Uni-Sign (Li et al., 2025) | 55.30 | 42.21 | 32.94 | 26.25 | 56.03 | 55.08 | 42.14 | 32.98 | 26.36 | 56.51 |
| SignLLM (Gong et al., 2024) | 42.45 | 26.88 | 17.90 | 12.23 | 39.18 | 39.55 | 28.13 | 20.07 | 15.75 | 39.91 |
| Sign2GPT (Wong et al., 2024) | - | - | - | - | - | 41.75 | 28.73 | 20.60 | 15.40 | 42.36 |
| **Sigma** | **56.83** | **44.09** | **34.94** | **28.24** | **58.04** | **55.97** | **43.00** | **33.91** | **27.30** | **57.58** |

in our SGT encoder, as shown in Table 20. The results show that both modalities independently support performance across ISLR, CSLR, and SLT tasks. Sign features slightly outperform text features on ISLR, highlighting their strength in capturing fine-grained visual details. In contrast, text features offer marginal gains on SLT, particularly in ROUGE-L, reflecting their advantage in encoding linguistic structure. The identical CSLR WER of 26.12 in both settings suggests that each modality provides similar semantic information for effective sequence alignment. These findings confirm the impact of our cross-modal pre-training strategy in learning semantically rich and transferable representations from both visual and textual sources.

## J COMPLETE RESULTS OF EXPERIMENTS

Due to the page limit of the main paper, certain experimental results could not be included. We present the full set of results (see Table 21, Table 22, and Table 23) here to ensure transparency and to support future research by providing comprehensive reference data.

## K LIMITATIONS

**1)** Despite the effectiveness of gloss annotations in improving ISLR and CSLR performance, their reliance presents a limitation. Annotating glosses, especially for large-scale datasets, is time-consuming and requires domain expertise to ensure accuracy and consistency. These glosses, even when annotated by experts, serve only as approximate representations of the corresponding sign sequences. While recent methods attempt to reduce or bypass gloss supervision, there is currently no best optimal solution that can fully replace gloss annotations without compromising performance for all aforementioned SLU tasks. As a result, existing SLU pipelines still depend heavily on manually curated glosses for training, which hinders scalability and limits applicability in low-resource or less-annotated sign languages.

**2)** While our framework unifies multiple SLU tasks, task-specific methods may still perform better in certain scenarios. In future work, we plan to incorporate additional modalities such as RGB and depth, which provide richer visual information and have the potential to further improve SLU performance. However, these modalities may introduce additional computational overhead that could impact real-time efficiency depending on the deployment context. We aim to explore balanced solutions that leverage the richness of multi-modal inputs while maintaining computational efficiency for real-time SLU.

## L    VISUALISING CROSS-MODAL ALIGNMENT WITH T-SNE

To visually evaluate the quality of the learned representations, we randomly sample eighty paired sign sequences and their corresponding translated text(s) to visualise the 2D distribution of sign and text features using **t-distributed stochastic neighbour embedding (t-SNE)**. This dimensionality reduction technique preserves local neighbourhood structure in high-dimensional data, making it especially effective for examining the alignment between modalities in the latent space. The features used in this analysis are derived from the downstream model of Sigma, ensuring that they reflect task-specific representations. Specifically, sign features are extracted from the output of the sign encoder, while text features are obtained immediately after the text embedding layer of the SGT decoder. This design choice reflects the decoding process during inference, where the sign encoder outputs are fed into the SGT decoder to generate textual representations. For each task, We compare the features learned by models initialised with only the mT5 weight to those initialised with pre-trained parameters from the pre-training stage. This comparison provides insight into how pre-training shapes cross-modal alignment and reveals the effectiveness of Sigma in learning semantically grounded representations.

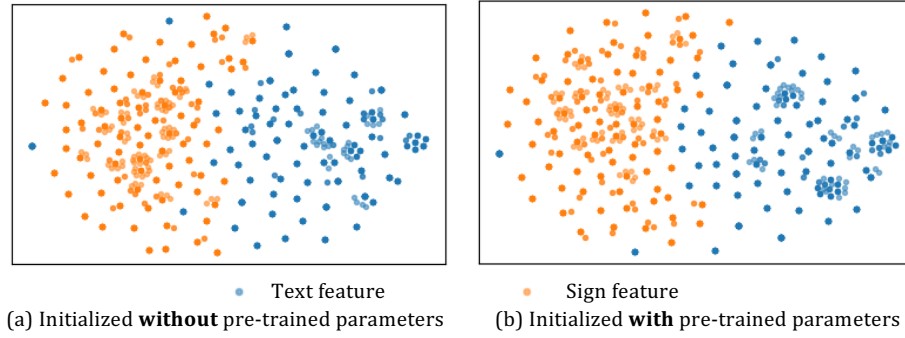

(a) Initialized **without** pre-trained parameters    (b) Initialized **with** pre-trained parameters

Figure 10: t-SNE visualisation of ISLR on WLASL.

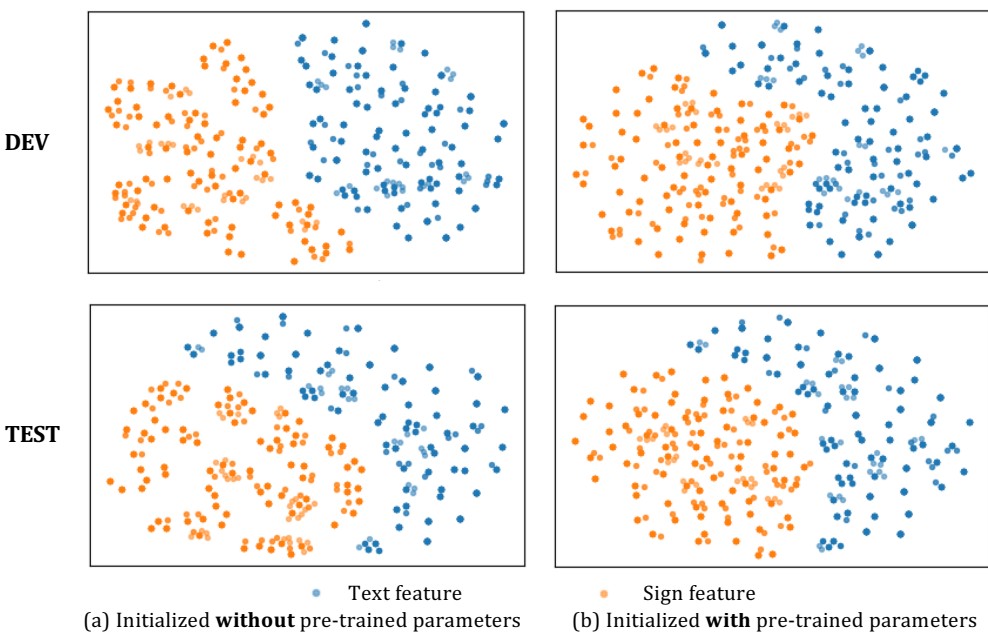

(a) Initialized **without** pre-trained parameters    (b) Initialized **with** pre-trained parameters

Figure 11: t-SNE visualisation of CSLR on CSL-Daily.

Figure 10 illustrates the effect of pre-training on cross-modal alignment in the ISLR task using the WLASL2000 dataset. The left panel shows a t-SNE visualisation of features from a model trained

from scratch, where sign and text features appear loosely scattered with blurred boundaries between modalities. This diffuse distribution reflects weak semantic alignment and limited interaction between visual and linguistic representations. In contrast, the right panel visualises features from the Sigma model initialised with pre-trained parameters. In this case, the sign features form tighter and more coherent clusters with clearer boundaries, demonstrating relatively stronger alignment with their corresponding textual representations. This comparison underscores the effectiveness of semantically informed pre-training in learning structured, discriminative representations that enhance cross-modal understanding in SLU.

The effect of pre-training on cross-modal alignment in the CSLR task is more pronounced than in the ISLR setting. As shown in Figure 11, we visualise t-SNE projections of sign and text features from the CSL-Daily dataset, comparing models with and without pre-trained initialisation. In both the DEV and TEST sets, the model trained from scratch produces dispersed and weakly aligned feature distributions across modalities. By contrast, the pre-trained model yields more compact, coherent clusters with noticeably improved alignment between sign and text features. This structural refinement highlights the effectiveness of semantically informed pre-training in strengthening visual-linguistic alignment, ultimately resulting in more robust and generalizable representations for CSLR.

Compared to ISLR and CSLR, the effect of pre-training in SLT is more evident than in ISLR but somewhat less so than in CSLR. Figure 12 presents t-SNE visualisations of the SLT task on the CSL-Daily dataset, contrasting feature distributions from models trained with and without pre-trained initialisation. In the absence of pre-training, the sign and text features appear loosely distributed with little structural alignment, reflecting weak semantic integration across modalities. By contrast, pre-trained models exhibit more compact clustering and clearer alignment between sign and text features in both the development and test sets. This suggests that pre-training not only enriches modality-specific semantics but also fosters cross-modal coherence essential for accurate and fluent translation. These findings reinforce the importance of semantically guided pre-training in shaping interpretable representations for SLT.

The t-SNE visualisations across ISLR, CSLR, and SLT consistently demonstrate that semantically informed pre-training effectively mitigates the cross-modality gap between sign and text representations. The degree of improvement varies, with the effect most pronounced in CSLR, moderately strong in SLT, and relatively limited in ISLR, the overall trend confirms that pre-training enhances both the semantic expressiveness of each modality and their alignment within a unified representation space.

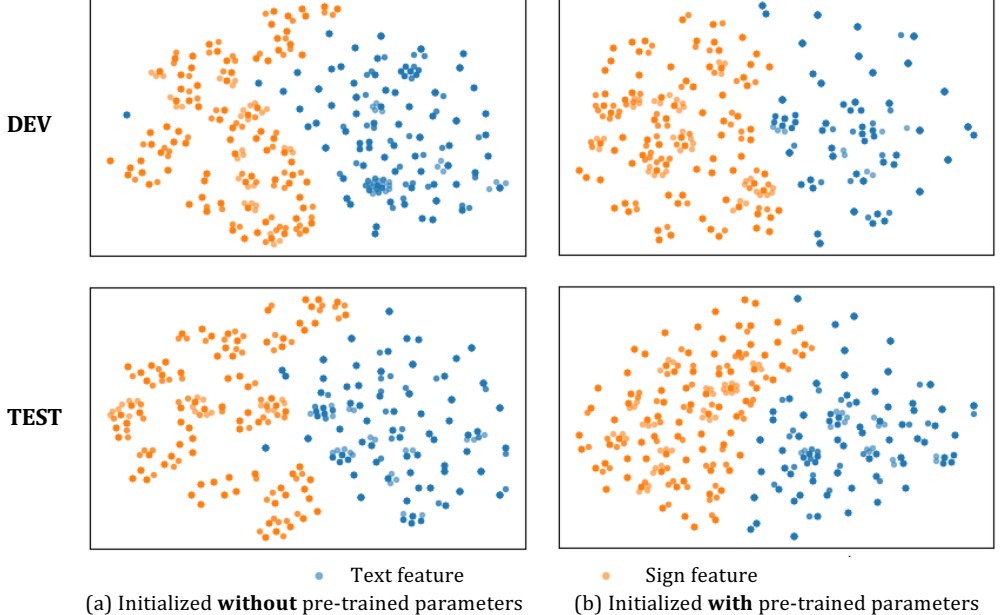

(a) Initialized **without** pre-trained parameters      (b) Initialized **with** pre-trained parameters

Figure 12: t-SNE visualization of SLT on CSL-Daily.

## M ADDITIONAL ABLATION STUDY

I In the main paper, we assess the impact of each core component by varying the number of fusion layers, balancing local-global feature modelling ($\alpha$), and adjusting the weights of text matching and language modelling ($\beta$). In this section, we extend those analyses with additional experiments targeting extreme configurations. Together with the t-SNE visualisations, these results provide further evidence that each component plays a valuable role in bridging the cross-modality gap. They also highlight that the effectiveness of pre-training depends on balanced feature modelling and structured modality interaction, which collectively ensure semantically aligned SLU representations across ISLR, CSLR, and SLT tasks.

Table 24: Impact of SignEF.

| Layers | WLASL2000 | | CSL-Daily | | | |
|---|---|---|---|---|---|---|
| | ISLR | | CSLR | | SLT | |
| | P-I↑ | P-C↑ | WER↓ | B@1↑ | B@4↑ | R@L↑ |
| **0** | **62.82** | **60.38** | **26.86** | **56.43** | **26.80** | **56.01** |
| 1 | 63.17 | 60.77 | 26.58 | 56.79 | 27.50 | 56.64 |
| 2 | 63.79 | 61.40 | 26.12 | 56.83 | 28.24 | 58.04 |
| 3 | 64.22 | 62.09 | 26.53 | 56.72 | 27.93 | 57.36 |
| 4 | 63.97 | 61.70 | 26.40 | 56.80 | 27.98 | 57.86 |
| 5 | 64.40 | 62.32 | 26.69 | 56.79 | 28.09 | 57.56 |

When the number of fusion layers is set to zero (see Table 24), the model disables the SignEF mechanism, leading to a performance drop across all SLU tasks. For instance, ISLR on the WLASL2000 dataset records its lowest performance in this setting, with per-instance and per-class accuracies of 62.82% and 60.38%, respectively. Similarly, CSLR on CSL-Daily yields a higher WER of 26.86%. SLT performance also declines, with BLEU-1 at 56.43, BLEU-4 at 26.80, and ROUGE-L at 56.01, indicating a broader degradation across tasks. These results indicate the importance of SignEF in differentiating visually similar gestures, especially in tasks like ISLR and SLT where semantic precision and expressive generation are crucial, which beyond what WER alone can capture. The absence of early visual-linguistic interaction hampers the ability of Sigma to establish strong alignment between sign and text representations. In contrast, even minimal integration of SignEF yields consistent gains, emphasising the importance of early-stage modality fusion for deeper semantic grounding and improved sequence modelling.

Table 25: Local-global feature balancing.

| Alpha | WLASL2000 | | CSL-Daily | | | |
|---|---|---|---|---|---|---|
| | ISLR | | CSLR | | SLT | |
| | P-I↑ | P-C↑ | WER↓ | B@1↑ | B@4↑ | R@L↑ |
| **0.0** | **63.85** | **61.72** | **27.12** | **56.00** | **27.16** | **57.32** |
| 0.2 | 64.14 | 62.14 | 26.52 | 56.11 | 27.99 | 57.89 |
| 0.4 | 64.30 | 62.24 | 26.55 | 56.65 | 28.12 | 57.82 |
| 0.5 | 64.40 | 62.32 | 26.12 | 56.83 | 28.24 | 58.04 |
| 0.6 | 64.14 | 62.09 | 27.05 | 56.82 | 27.99 | 58.10 |
| 0.8 | 64.14 | 62.08 | 26.65 | 56.21 | 27.66 | 57.65 |
| **1.0** | **63.67** | **61.55** | **27.27** | **55.99** | **27.00** | **57.24** |

The performance at the two extremes of the $\alpha$ parameter (0.0 and 1.0) highlights the importance of balancing local and global feature alignment (see Table 25). When $\alpha$ is set to 0.0, the model relies solely on global alignment, ignoring fine-grained local interactions. This leads to degraded performance across all SLU tasks. On the other hand, when $\alpha$ is set to 1.0, the model depends entirely on local contrastive learning, without leveraging global class-level alignment. Comparing these extremes reveals that ISLR and SLT performance suffers more when relying exclusively on local features, while CSLR shows slightly better results under local-only alignment. These findings suggest that neither local nor global alignment alone is sufficient to capture the full spectrum of semantic relationships required for SLU. The strongest performance is achieved when both are combined,

Table 26: Trade-off analysis between text matching and language modelling.

| Beta | WLASL2000 | | CSL-Daily | | | |
|---|---|---|---|---|---|---|
| | ISLR | | CSLR | | SLT | |
| | P-I↑ | P-C↑ | WER↓ | B@1↑ | B@4↑ | R@L↑ |
| **0.0** | **64.07** | **61.82** | **26.77** | **56.01** | **27.63** | **57.36** |
| 0.2 | 64.30 | 62.21 | 26.62 | 56.31 | 27.82 | 57.88 |
| 0.4 | 64.40 | 62.15 | 26.43 | 56.38 | 27.88 | 57.76 |
| 0.5 | 64.27 | 62.03 | 26.12 | 56.83 | 28.24 | 58.04 |
| 0.6 | 64.40 | 62.32 | 26.33 | 56.00 | 27.79 | 57.43 |
| 0.8 | 64.12 | 62.22 | 26.27 | 55.58 | 27.72 | 57.36 |
| **1.0** | **63.92** | **61.76** | **26.55** | **55.26** | **26.98** | **56.88** |

reinforcing the necessity of joint local-global modelling for learning semantically meaningful and generalizable representations across diverse SLU tasks.

The results for $\beta = 0.0$ and $\beta = 1.0$ in Table 26 illustrate the importance of balancing text matching and language modelling during training. At $\beta = 0.0$, when the objective is driven entirely by text matching, performance drops across all SLU tasks. At the other end, with $\beta = 1.0$, the model relies only on language modelling, yielding lower SLT results than at $\beta = 0.0$, but more pronounced degradation in ISLR and similar degree of degradation in CSLR. These observations show that placing too much emphasis on either modality-specific alignment or generative fluency fails to deliver consistent performance across tasks. The performance drop at both ends highlights the need for a balanced approach, where integrating text matching and language modelling allows the model to align cross-modal semantics while producing coherent textual outputs.

The extent of performance degradation under extreme parameter settings in our proposed method varies across tasks and evaluation metrics. Overall, the most substantial decline occurs when the SignEF module is removed, underscoring its essential role in enabling effective cross-modal interaction. One exception is the BLEU-1 score, where the decrease is relatively modest, suggesting that surface-level lexical matching may be less dependent on early fusion compared to deeper semantic metrics like BLEU-4 or ROUGE-L. When comparing the impact of imbalanced feature modelling versus unbalanced text matching and language modelling, the former tends to result in more pronounced performance drops. An exception occurs when $\beta = 1.0$, where BLEU-4 reaches its second lowest value, indicating that overreliance on language modelling can impair the ability of the model to preserve fine-grained sign-text correspondence. These findings highlight the need for balanced integration across both feature learning and text objectives to achieve robust and semantically aligned SLU performance.

## N    ETHICS STATEMENT

Our work focuses on sign language understanding, aiming to improve accessibility and communication for people with hearing or speech impairment. The datasets we use (WLASL, CSL-Daily, How2Sign, OpenASL) are publicly available and widely adopted in SLU research. We strictly follow dataset licenses and use them only for academic purposes. No personally identifiable information or sensitive attributes beyond the original releases are introduced. We acknowledge the cultural and linguistic importance of sign languages and stress that our models are intended to support accessibility rather than replace human interpreters.

## O    REPRODUCIBILITY STATEMENT

We ensure reproducibility by providing detailed descriptions of our models, objectives, and training configurations in the main text and appendix. Dataset statistics and preprocessing steps are clearly reported. Hyperparameters, loss formulations, and evaluation protocols are included to enable replication. To further support reproducibility, we will release our code upon publication. These resources will allow the community to replicate our experiments and extend our work on sign language understanding.

## P    USE OF LARGE LANGUAGE MODELS

We used a large language model (ChatGPT) as an assistive tool to polish certain parts of the paper for clarity and readability. Specifically, we employed prompts such as "Polish the following" to improve the fluency and presentation of text that had already been drafted by the authors. ChatGPT was not used for ideation, analysis, experiment design, or the generation of technical content. All research ideas, methods, experiments, and results were conceived, implemented, and validated by the authors. We take full responsibility for the content of this work, and ChatGPT is not considered a contributor or author.

