# OpenReview forum: "Sigma: Semantically Informative Pre-training for Skeleton-based Sign Language Understanding"
_ICLR.cc/2026/Conference — Submitted to ICLR 2026_

### Official Review · Reviewer_EVSg · 2025-10-20

**Soundness:** 3
**Presentation:** 4
**Contribution:** 3
**Rating:** 8
**Confidence:** 3

**Summary:**

The authors present a model and framework for sign language recognition that tackles three challenges in the current scientific literature in this domain. They ground skeletal data in textual contexts, they balance local and global context critical for sign language understanding, and they align different modalities. An early fusion method based on cross-attention allows the model to learn from visual encoding and text modalities, and alignment is learned to maximize agreement between these modalities.

Using as little as 10 pre-training and 10 fine-tuning epochs, the authors are able to achieve competitive results on WLASL2000 and CSL-Daily.

In summary, this paper presents a new method for pre-training and fine-tuning models for ISLR, CSLR, and SLT, showing the importance of cross-modal training.

**Strengths:**

The paper is clearly situated within a relevant body of work, and the necessary background knowledge is explained in sufficient detail.
The overall presentation of the paper is good: it is well structured and explains the relevant aspects. The tables and figures in the table support the main text well. The design choices made by the authors are clearly grounded and informed.

Content-wise, the paper tackles an important limitation of recent work in SLR and SLT, that is: that most models incorporate only visual cues and neglect the linguistic aspects of the task at hand. By including a model and alignment for global context and text, these authors advance the state-of-the-art in these domains.

**Weaknesses:**

The paper is slightly lacking in terms of how to apply this model. In particular, it is unclear to me how this model can be used for ISLR, CSLR, and SLT. See also: "Questions".

The paper introduces three new methods at the same time. This sometimes makes it difficult to follow. However, at the end, the impact of the individual contributions is measured and reported.

The paper could also benefit from expanding the related works section to include a wider array of papers. Currently, the scientific literature overview is somewhat limited. For example, more excellent work has been done on keypoint-based SLR and SLT that is not mentioned here.

**Questions:**

I would consider using SLPT instead of SLP as the abbreviation for Sign Language Pre-training, as SLP is often used for Sign Language Production.

Please check your citation commands, as references appear in the text as "Author (Year)" even when "(Author, Year)" would have been appropriate.

Can you elaborate on how your model is used at inference time? During fine-tuning, textual inputs are used. Can these be discarded for inference? If the SGT decoder is removed from the architecture, how are logits predicted for ISLR/CSLR and how are tokens generated for SLT?

---

> ### Author Response · Authors · 2025-11-20
>
> We sincerely thank you for the thoughtful and encouraging review. We appreciate your recognition of the clarity of our presentation, the coherence of our design choices, and the importance of the linguistic aspects in SLU. Below, we respond to your questions in detail.
>
> **1. Unified processing of ISLR, CSLR, and SLT in training and inference.**
>
> Sigma is designed to provide a single, unified solution for ISLR, CSLR, and SLT. Rather than treating recognition and translation as fundamentally different tasks (classification versus generation), Sigma casts all three tasks as conditional language modelling. Given a sequence of visual features derived from skeletal data, the model predicts a sequence of tokens.
>
> The mT5 decoder provides a flexible output space. For ISLR, the “sentence” to be predicted consists of one or a few label tokens. For CSLR, it becomes a gloss sequence. For SLT, it becomes a language translation. During training, the procedure remains identical across tasks; the only difference lies in the textual target provided. During inference, Sigma requires no textual input. The encoder processes the skeletal sequence, produces visual features, and the decoder autoregressively generates the appropriate output tokens. Depending on the task, these tokens can be interpreted as a textual class label, a gloss sequence, or a translated sentence.
>
> In this way, the same architecture, the same training loop, and the same inference pathway seamlessly serve all three SLU tasks. This is one of the key motivations and advantages of Sigma: it provides a unified, scalable formulation rather than maintaining three separate task-specific pipelines.
>
> **2. How are logits predicted when the SGT decoder is removed?**
>
> In Sigma, we do not remove the SGT decoder for any of the three tasks. Instead, we unify ISLR, CSLR, and SLT under a single formulation: all tasks are treated as conditional language modelling. This means that single-class classification (ISLR), multi-class classification (CSLR), and full-sentence translation (SLT) are all expressed as token-generation problems. The decoder, therefore remains essential, as it is responsible for producing the output token sequence regressively for every task.
>
> For ISLR, the decoder generates a short sequence that corresponds to the textual class label. For CSLR, it generates a gloss sequence. For SLT, it generates a natural language sentence.
>
> Since all outputs are obtained through the same autoregressive decoding mechanism, the logits are always produced by the mT5 decoder’s language modelling head. There is no need for a task-specific classifier head, and no setting in which the decoder is removed. This unified formulation is precisely what allows Sigma to serve all three SLU tasks with the same architecture and inference process.
>
> **3. Refining citation format, expanding related work, and choosing clearer abbreviations**
>
> We sincerely appreciate these helpful suggestions. The citation formatting has been revised for consistency and clarity. We have added additional skeleton-based SLU literature to further enrich the related works in Appendix B. To ensure unambiguous terminology, we now use the term “SLPT” for sign language pre-training, clearly distinguishing it from sign language production and avoiding any potential confusion.

---

> > ### Comment · Reviewer_EVSg · 2025-11-21
> >
> > Thank you for the clarifications.
> >
> > > Sigma is designed to provide a single, unified solution for ISLR, CSLR, and SLT. Rather than treating recognition and translation as fundamentally different tasks (classification versus generation), Sigma casts all three tasks as conditional language modelling
> >
> > This makes it all clear to me. I recommend adding/highlighting these few sentences in the paper to make sure that the message gets across to all readers.
> >
> > With this, all of my questions have been properly addressed.

---

> > > ### Author Response · Authors · 2025-11-23
> > >
> > > Thank you for the follow-up.
> > >
> > > We have updated the manuscript to clearly state that Sigma provides a single unified formulation for ISLR, CSLR, and SLT by treating all three as conditional language modelling in Section 3.3 of the main paper.
> > >
> > > We appreciate your careful reading and constructive guidance.

---

### Official Review · Reviewer_Kc3q · 2025-10-31

**Soundness:** 3
**Presentation:** 3
**Contribution:** 3
**Rating:** 4
**Confidence:** 4

**Summary:**

The paper proposes Sigma, a unified, skeleton-based pre-training framework for Sign Language Understanding (SLU). It aims to solve three key limitations in existing methods: weak semantic grounding, where models fail to connect low-level motion patterns to linguistic meaning ; an imbalance between modeling fine-grained local details and high-level global context ; and inefficient cross-modal learning

**Strengths:**

1. Unified framework: this paper presents a single, unified framework that is fine-tuned to achieve state-of-the-art performance on all three major SLU tasks (ISLR, CSLR, and SLT), demonstrating its versatility and strong generalization capabilities.

2. Architectural Novelty: The proposed components (SignEF and HAL) are not arbitrary but are specifically designed to solve clearly articulated problems in SLU, namely the local-global imbalance and weak semantic grounding.

3. Strong results: Sigma achieves new state-of-the-art results on multiple diverse benchmarks, including WLASL2000, CSL-Daily, How2Sign, and OpenASL, surpassing previous methods in recognition accuracy, WER, and translation (BLEU/ROUGE) scores.

4. Efficiency: this work effectively utilizes skeleton-based data, which is robust to background and subject variations, requires significantly less storage, and is more computationally efficient than RGB-based methods, making it a more scalable solution.

**Weaknesses:**

1. The expression of Eq.2 is not clear. For example, what's the meaning of n and k here? Additionally, s and c are not defined here. This hinders the readers from understanding the meanings.
2. The intuition of dividing works into sub-word level units is strange. As stated in the paper, “curiosity” is split into“curios” and “ity,”. However, "curios" doesn't have any concrete meaning in English, and in sign language the actor doesn't express meanings in a sub-word level. The recombination of different sub-word results in misclarification of the meaning which is hardly understood by human beings, let alone aligning with visual features expressed by the actor. Thus, in the perspective of aligning texts with visual features, the alignment should be performed in the word level instead of the sub-word level. The effect of this design is also not ablated in the experiments.
3. The experssion of Algorithm 1 is not clear. In line 7, performing softmax operation will not diminish the dimension L. Besides, the dimension of performing summation in line 8 seems wrong. Besides, does the Cluster-wise sign-to-text similarity have a ground-truth for supervision? Or just enforce it to approximate 1.0?
4. In HAL, why omitting temperature scaling could be stated as fine-grained local alignment?
5. The design of SGT encoder is borrowed directly from previous works.
6. The effects of different components are not ablated in the experiments, e.g. the effects of HAL, early fusion and so on.

**Questions:**

See above. I will adjust my score according to the authors response.

---

> ### Author Response · Authors · 2025-11-20
>
> We appreciate your acknowledgement of the effectiveness of Sigma, and the strong generalisation of our unified framework. Your feedback gave us an opportunity to reflect on our design choices and articulate clearly the reasoning and insights that shaped our method.
>
> **1.  Clarification of global and local cluster-wise similarity used for HAL in Equation (2)**
>
> Equation (2) defines two ways of measuring the similarity (global and local cluster-wise similarity) between sign embeddings and text embeddings used for hierarchical alignment learning. The full definitions are provided in Lines 230–246. In the revised manuscript, we have clarified these formulations to improve readability and ensure that the roles of both similarity terms are easy to follow.
>
> The similarity function is:
>
> $M ^{x }_{s2t} = \operatorname{sim}(S _{r},\ T _{j}; \phi)$
>
> where the superscript $x$ determines whether we compute global sign-to-text similarity $M^{g}_{s2t}$ or local sign-to-text similarity $M ^{l} _{s2t}$.
>
> **1.1. Global similarity $(x=g)$**
>
> $M ^{g} _{s2t} = g _{s}(s _{\mathrm{cls}}) ^{\top}g _{t}(t _{\mathrm{cls}})$
>
> This computes the dot-product similarity between:
> - $s _\mathrm{cls}$: the class token from the sign encoder,
> - $t _\mathrm{cls}$: the class token from the text encoder.
>
> After both tokens are projected into the shared embedding space using the projection heads $g _{s}$ and  $g _{t}$, their cosine-like similarity is used to represent the holistic, sentence-level semantic alignment. Global alignment captures coarse-grained semantic correspondence between the entire sign sequence and the full text.
>
> **1.2. Local similarity $(x=l)$**
>
> $M ^{l} _{s2t}
> = \sum _{i=1} ^{n} \max _{j \in \{1,\ldots,k\}}
> \big(g _{s}(s _{i}) ^{\top}\, g _{t}(c _{j}) \big)$
>
> This equation measures fine-grained local cluster-wise alignment between sign representations and text clusters.
>
> Here is what each component means:
>
> **Sign side**
> - The sign encoder outputs $n$ local sign tokens $s _{1}, s _{2}, \ldots, s _{n}$.
> - Each token represents the motion extracted for one frame of the sign video.
>
> **Text side**
> - The text tokens are aggregated by our cluster aggregator into $k$ text clusters $c _{1}, c _{2}, \ldots, c _{k}$.
> - Each cluster corresponds to a semantically coherent group of token embeddings (a soft proxy for gloss-like semantic units).
>
> This procedure ensures that each local sign motion finds its best semantic match among the text clusters. Local alignment encourages the sign encoder to encode meaningful sub-motion patterns that correspond to textual semantics.
>
> **2. Clarification of Algorithm 1 for cluster-wise sign-to-text similarity**
>
> **2.1 Regarding the dimensionality after the softmax operation:**
>
> The vector $R \in \mathbb{R} ^{B \times N _b}$ stores the maximum cluster similarity for each sign token, and  $w = \mathrm{softmax}(R)$ produces a normalised weight distribution over these sign tokens, reflecting their relative importance for local alignment. The dimension of $w$, therefore, remains $B \times N _b$.
>
> **2.2 Regarding the summation dimension:**
>
> The summation is performed over the sequence dimension of the sign tokens:
>
> score $= \sum _{\mathrm{dim}=2} (w \odot R )$
>
> which reduces the output of row-wise operation $R \in \mathbb{R} ^{B \times N _b}$ to a vector of length $B$, where each entry of the score represents the list of similarities for a batch of samples. $\mathrm{score}[b]$ or $M ^{l} _{s2t}[b][b]$ is expected to be higher than all other similarity values across the dimension of the batch. The notation has now been refined in the revision to avoid ambiguity. We also improve the visualisation of this Algorithm in Figure 9.
>
> **2.3 Regarding supervision of the cluster-wise similarity:**
>
> The cluster-wise sign-to-text similarity does not have explicit ground truth labels. Instead, it serves as a self-supervised soft constraint that encourages each sign feature to be close to the most semantically relevant text clusters. The objective does not enforce the score to be 1.0; rather, it encourages relative alignment within a shared embedding space. This is similar in spirit to soft alignment used in prior work on contrastive and cross-modal representation learning (CLIP).

---

> > ### Author Response · Authors · 2025-11-20
> >
> > **3. Clarification of subword-level textual tokens produced by the mT5 tokenizer**
> >
> > We appreciate the concern and agree that sub-word units such as “curios” and “ity” do not carry standalone semantic meaning for human readers, nor do signers express meaning at this granularity. We clarify that the use of sub-word units in our method is not a design choice specific to Sigma, but rather arises from the mT5 tokenizer used by the textual backbone. mT5 employs a tokenizer trained on multilingual corpora, and its segmentation decisions are inherited from the backbone rather than introduced by our framework. Our cluster aggregator is designed to naturally group related sub-word pieces into coherent semantic units, effectively reconstructing meaningful word-level representations, and to form gloss-like semantic units. This process restores linguistically interpretable alignment targets and reduces the noise that would arise from matching signs to incomplete fragments.
> >
> > **4. Temperature scaling in HAL**
> >
> > We omit temperature scaling in the local contrastive branch because it focuses on fine-grained, high-variance local distributions, where a fixed temperature can oversmooth gradients and obscure sharp local correspondences. In contrast, the global branch uses temperature to stabilise holistic alignment.
> >
> > **5. SGT encoder and SGT decoder**
> >
> > This dual objective (matching and generation) of the SGT encoder gives Sigma the capacity to meet the demands of both recognition-oriented tasks, including ISLR and CSLR, and generation-oriented tasks, including SLT. We bring these three tasks into a single, coherent formulation by treating them all as conditional language modelling. In this view, single-label prediction in ISLR, gloss-sequence prediction in CSLR, and full-sentence translation in SLT all become token-generation problems. The encoder plays a central role: during pre-training, it learns to align signs with text while continuously producing token sequences, which equips it to be reused for downstream decoding. This reuse of a single encoder–decoder pathway, rather than switching between different task-specific heads, is new in SLU.
> >
> > **6. Component ablations**
> >
> > Ablations isolating SignEF, HAL, and SGT are provided in Table 2-4 and Appendix I. We appreciate this suggestion and have linked them for better readability.

---

> > > ### Comment · Reviewer_Kc3q · 2025-11-25
> > > **Response**
> > >
> > > I have read the author rebuttal. My concerns are now as follows. (1) My most concerns fall in limited novelty in my view. The major components of the paper is borrowed directly from previous works. For example, sign-aware early fusion is a cross-attention operation. Hierarchical alignment learning is derived from constrasive learning. The SGT encoder adopts sign-text matching and language modeling which is adopted from BLIP2. This work successfully transfers these component into the SLU  task to build a unified model. (2) Ablations are insufficient. Tab.2-4 only ablate the hyper-parameters of each module, but don't test the effectiveness of each module, like what if there modules are concalled or replaced as others? (3) The reason of omitting temperature scaling could be stated as fine-grained local alignment doesn't convince me. The temperature scaling factor controlls the shape of feature distribution but is not related local or global features. Overall, for me, the weakness of this paper lies in partially limited methodology novelty and unclear explanations for certain algorithms (like dividing works into sub-word level units for contrasive learning). The strength lies in strong performance and a unified framework. I would keep my score while accepting this paper is somewhat acceptable.

---

> ### Author Response · Authors · 2025-11-27
>
> We sincerely thank the reviewer for the thoughtful comments. We appreciate that you recognised the strengths of Sigma in terms of unified design, strong performance, and the potential impact on SLU. Below, we address your remaining concerns regarding novelty, ablations, temperature scaling, and the use of sub-word units.
>
> **1. Clarifying our novelty and the insight behind Sigma beyond prior skeleton-based SLU methods**
>
> Prior work typically treats isolated sign language recognition (ISLR), continuous sign language recognition (CSLR), and sign language translation (SLT) as three separate classification and generation tasks, which leads to disconnected objectives and prevents cross-task knowledge transfer. In contrast, Sigma integrates these tasks within a single pre-training framework. Cross-attention alone does not explain our design. We adapt it so that sign features can gradually absorb the structural patterns learned by the text encoder. This early interaction lets the sign encoder inherit useful linguistic organisation from the very beginning, guiding raw kinematic trajectories toward semantically meaningful representations as they form. Such early coupling is important for improving semantic grounding, a challenge that skeleton-based SLU models have long struggled with. While contrastive learning is promising for multimodal alignment, its effectiveness in SLU is constrained when used in its conventional form. Shallow cross-modal interaction between low-level kinematics and high-level semantics, together with coarse and global objectives, is insufficient to establish meaningful semantic grounding. Under the limited data typical of SLU benchmarks, such signals dissipate quickly, leaving the model with weak or unstable grounding. Sigma strengthens this process by guiding the sign encoder with linguistic structure early and refining alignment through hierarchical global and cluster-wise interaction. While combining matching and generation losses has precedent in vision-language pre-training, Sigma integrates them into a unified formulation tailored to SLU. By treating ISLR, CSLR, and SLT all as conditional language modelling, the same linguistic decoder becomes responsible for both recognition and translation. In this formulation, isolated labels, gloss sequences, and full sentences all emerge from a single autoregressive decoding pathway. This shared pathway has not appeared in prior SLU approaches and is what enables Sigma to handle the three mainstream tasks within one coherent framework.
>
> **2. Component ablations**
>
> We have linked the additional experiments presented in Appendix L to the main paper, where we ablate the use of our sub-modules for the unified method. These results, included in the revision, demonstrate that each module is necessary for achieving strong performance across the mainstream SLU tasks.
>
> **3. Temperature scaling**
>
> Local alignment operates on fine-grained, high-variance sign features. Introducing a temperature parameter in this branch flattens similarity distributions and weakens sensitivity to precise sign-cluster correspondences. For this reason, we omit temperature in the local branch but keep it in the global branch, where distributions are more stable.
>
> **4. Sub-word units**
>
> We clarified that Sigma does not design or choose sub-word segmentation; it follows the tokenizer built into mT5. To handle the lack of linguistic meaning in fragments like “curios” and “ity,” HAL’s cluster aggregator groups sub-words into coherent semantic clusters that serve as gloss-like alignment units. This reduces noise and improves sign-text matching.

---

### Official Review · Reviewer_LWyo · 2025-11-02

**Soundness:** 2
**Presentation:** 3
**Contribution:** 2
**Rating:** 4
**Confidence:** 5

**Summary:**

The proposed Sigma framework offers a predictable combination of early fusion, hierarchical alignment, and multi-task pre-training—techniques already commonplace in multimodal learning. While it claims to address known SLU limitations (weak semantics, local-global imbalance, cross-modal gaps), the solutions lack novelty and largely rehash standard contrastive and alignment strategies.

**Strengths:**

The sole merit of Sigma lies in its marginal SOTA improvements across recognition and translation benchmarks. These gains—while consistent—are small and stem from routine integration of early fusion, hierarchical alignment, and multi-objective pre-training, not from conceptual innovation.

**Weaknesses:**

In a mature field, such incremental performance uplifts, absent methodological novelty or deeper insight, offer limited scientific value. The work is technically sound but contributes little beyond leaderboard chasing.

**Questions:**

(1) Skeleton inputs are de facto standard in SLU. Beyond re-emphasizing robustness to appearance, what specific insight or technical leap distinguishes Sigma from the dozens of prior skeleton-based models?

(2) The paper claims "sign-aware early fusion" resolves weak semantic grounding. This connection is not intuitive—fusing modalities earlier does not inherently bridge motion patterns to linguistic meaning. Provide concrete evidence (e.g., ablation on semantic alignment metrics) showing how this mechanism grounds low-level kinematics in high-level semantics.

(3) Are the proposed modules (early fusion, cluster-based hierarchical aggregator, etc.) plug-and-play improvements for any skeleton-based SLU backbone, or are the gains architecture-dependent? Without cross-model validation, claims of broad applicability remain unsubstantiated.

---

> ### Author Response · Authors · 2025-11-20
>
> We sincerely thank you for taking the time to review our work. Our goal is to build a unified SLU model to enhance the quality of life for around 70 million people with hearing or speech impairments. We appreciate your acknowledgement of the technical sounds of the framework and the SOTA performance it consistently achieved. Your feedback has given us the opportunity to further clarify key aspects of our framework, particularly in terms of its novelty, effectiveness, and practical considerations.
>
> **1. Clarifying our novelty and the insight behind sigma beyond prior skeleton-based SLU methods**
>
> Sigma differs from existing skeleton-based SLU models by offering a unified solution for mainstream SLU tasks. Prior skeleton-based SLU work typically treats isolated sign language recognition (ISLR), continuous sign language recognition (CSLR), and sign language translation (SLT) as three separate classification and generation tasks, which leads to disconnected objectives and prevents cross-task knowledge transfer. In contrast, Sigma integrates these tasks within a single pre-training framework. While contrastive learning is promising for multimodal alignment, its effectiveness in SLU is constrained when used in its conventional form. Shallow cross-modal interaction between low-level kinematics and high-level semantics, together with coarse and global objectives, is insufficient to establish meaningful semantic grounding, especially under the limited data available in SLU benchmarks. Our method focuses on strengthening this grounding. Initially, by inheriting from a pre-trained machine translation model, Sigma benefits from a shared space that is broader and has more structured linguistic knowledge than what is available from sign language datasets alone, enriching the textual side with mature lexical and syntactic priors. Sign-aware early fusion
> further encourages the sign encoder to access this rich feature space at the initial stages of representation learning, letting motion trajectories and body articulations relate to linguistically relevant textual cues early and gradually transform from raw kinematic signals into semantically grounded representations. This prepares the model for deeper alignment that shallow contrastive objectives could not reach, allowing motion cues to settle naturally into the linguistic structure inherited by the sign encoder rather than drifting away and suffering from large-scale forgetting. The proposed hierarchical alignment learning further refines this process by combining holistic alignment with cluster-wise interactions that approximate gloss-like semantic units without requiring gloss annotation. This design provides a more faithful bridge between continuous sign motions and the latent linguistic structures they express. Finally, the SGT encoder balances the needs of both classification-based tasks (ISLR & CSLR) and generation-based tasks (SLT), enabling Sigma to function as a unified SLU framework. Together, these components form a coherent system, enabling skeleton-based models to benefit from linguistic priors and structured alignment in ways that conventional task-specific approaches do not.

---

> ### Author Response · Authors · 2025-11-20
>
> **2. Empirical evidence of enhanced semantic grounding**
>
> Unlike generic human actions, sign languages operate as full natural languages. Efficient semantic grounding in visual data remains one of the central challenges in SLU. The proposed Sigma is aimed at offering an effective solution to improve semantic grounding, rather than relying solely on either of the proposed components on its own. The training of Sigma starts by taking steps into a semantic space inherited from the pre-trained machine translation model (mT5). This semantic space is far richer and more structured than what sign language data alone can offer. This brings mature lexical and syntactic knowledge into the learning process and provides a stable linguistic anchor for visual representations. Sign-aware early fusion adjusts the sign encoder with the true text feature that corresponds to the visual feature in a shared semantic space at the earliest stage of representation learning. As the visual features flow into Sigma, they soon encounter linguistically meaningful cues, and the raw motion trajectories begin to shift and reorganise in response. Over the course of training, the visual stream gradually transforms from a collection of kinematic signals into representations that carry semantic intent. This staged interaction prepares the model for deeper alignment and empowers Sigma to overcome the limitations of shallow, purely contrastive objectives, ultimately producing visual features that “understand” the language they are aligned with in a better sense.
>
> To further assess whether Sigma improves the alignment between low-level kinematics and high-level semantics, we compute Centered Kernel Alignment (CKA) between visual features and the corresponding textual features. CKA is a widely used metric for quantifying representation similarity across modalities. A higher CKA value indicates better cross-modal correspondence.
>
> CKA is computed as follows:
>
> $\tilde{S}=S-\frac{1}{N _s} \sum _{i=1} ^{N _s} S _i,
> \qquad$
> $\tilde{T}=T-\frac{1}{N _t} \sum _{i=1} ^{N _t} T _i .$
>
> CKA$(\tilde{S}, \tilde{T}) =
> \frac{\left\lVert \tilde{S}^\top \tilde{T} \right\rVert_F^2}
> {\sqrt{\left\lVert \tilde{S}^\top \tilde{S} \right\rVert_F^2 \cdot
>        \left\lVert \tilde{T}^\top \tilde{T} \right\rVert_F^2}}$
>
> CKA measures the similarity between sign and text features after centering. $S \in \mathbb{R} ^{B \times D}$ and $T \in \mathbb{R} ^{B \times D}$ represent the pooled features for sign and text obtained by averaging over the sequence dimension, where $B$ indicates the batch size and $D$ indicates feature dimension.
>
> Sign features are extracted from the output of the sign encoder, while text features are obtained immediately after the text embedding layer of the SGT decoder. This design choice reflects the decoding process during inference, where the sign encoder outputs are fed into the SGT decoder to generate textual representations. We compare the features learned by models initialised with only the mT5 weight to those initialised with pre-trained parameters from the pre-training stage in the table below. This comparison provides insight into how our methods shape cross-modal alignment and reveals the effectiveness of Sigma in learning semantically grounded representations (We follow the same feature extraction procedure used to generate the t-SNE visualisation shown in Appendix L).
>
> As shown in the table below, Sigma’s pre-training substantially increases cross-modal CKA, demonstrating that the sign-text representations are more semantically aligned:
>
> Table. Sign-text CKA on CSL-Daily, along with BLUE 4 scores
>
> |with Sigma's pre-training|DEV-B@4|DEV-CKA(%)|TEST-B@4|TEST-CKA(%)|
> |-|-:|-:|-:|-:|
> |No|0.12|46.32|0.05|46.55|
> |Yes|28.24|66.02|27.30|67.37|
>
> This analysis reveals three key observations:
> 1. Sigma’s pre-training yields considerable improvement in representation similarity. CKA increases from roughly 46% to 66-67%, a substantial gain of around 20 percentage points, reflecting much stronger sign-text semantic alignment.
> 2. The improvement in CKA correlates with actual task performance. BLEU 4 improves from near-zero to 27-28, indicating that the strengthened cross-modal alignment indeed supports downstream translation, moving beyond superficial interactions.
> 3. Without Sigma’s pre-training, semantic grounding remains weak.
>
> These results provide direct, quantitative evidence that our method contributes to improved semantic grounding. The increase in CKA and downstream BLEU demonstrates that our work bridges low-level motion patterns and high-level linguistic meaning in a better sense.

---

> ### Author Response · Authors · 2025-11-20
>
> **3.  Architectural generality of the proposed modules**
>
> The backbone mT5 model can be very easily updated. Our choice of it reflects the practical constraints of available computational resources and the scale of current sign language datasets. Sigma is designed so that larger or more capable language backbones can be plugged in seamlessly, allowing the model to grow as more data and resources become available.

---

### Official Review · Reviewer_q949 · 2025-11-03

**Soundness:** 3
**Presentation:** 3
**Contribution:** 3
**Rating:** 6
**Confidence:** 3

**Summary:**

The paper presents Sigma, a unified pretraining framework for sign language understanding (SLU), designed to address the long-standing challenge of developing scalable and generalisable models that perform effectively across multiple SLU tasks. To this end, the authors introduce SignEF, HAL, and SGT, which collectively tackle the challenge and deliver impressive results across several benchmark evaluations.

**Strengths:**

The paper presents a unified pre-training framework for skeleton-based sign language understanding that integrates sign-aware early fusion (SignEF), hierarchical alignment learning (HAL), and multi-task pre-training within a single architecture. The design of SignEF is particularly novel, as previous studies in sign language research have seldom explored the potential benefits of cross-modal interaction, typically adopting architectures that fuse features only after unimodal extraction. In contrast, SignEF introduces a new and promising direction for future research. Moreover, the introduction of a local cluster-wise contrastive learning mechanism is also innovative. By employing a cluster aggregator to integrate features, this mechanism compresses feature representations and better aligns the sign and text spaces, leading to improved performance. Overall, this paper provides a creative perspective on integrating pose information with linguistic context, demonstrating that pose data alone can serve as a powerful foundation for SLU.

The paper is technically sound and supported by extensive empirical validation. A series of ablation studies thoroughly examine the effects of core components, benefiting from comprehensive experimental results and a simple architectural design. These ablations provide transparent justification for the architectural choices and enhance reproducibility. Moreover, the experiments cover multiple benchmarks and downstream tasks (ISLR, CSLR, SLT), demonstrating consistent performance gains across all settings.

**Weaknesses:**

1. Regarding the choice of the pre-trained language model, while the paper employs mT5 Base as the textual backbone, a relatively modest choice given the availability of larger mT5 variants, it would be beneficial to clarify the motivation behind this selection and whether Sigma can be readily extended to larger backbones to enhance reproducibility and future extensions.

2. It would be informative to report the additional computational and memory costs incurred by SignEF and HAL compared with the baseline.

3. The overall structure of the manuscript is well organized; however, certain bold and italicized expressions appear somewhat unconventional and could be refined to enhance the readability and stylistic consistency of the text. For instance, the references to SignEF and HAL in Lines 160–161, as well as WLASL2000 in the header of Table 18, may be appropriately revised.

4. Sigma is built exclusively around pose data, which the authors highlight as “clean and privacy-preserving”. However, this design choice completely excludes the rich RGB information, which could result in the loss of important visual cues. Could the authors further discuss when RGB information becomes particularly important?

5. It would be helpful if the authors could discuss the scalability and multilingual ability of Sigma.

**Questions:**

Please refer to the Weakness section above.

---

> ### Author Response · Authors · 2025-11-20
>
> We sincerely thank you for the thoughtful and encouraging review. We appreciate your recognition of the novelty of our method, the depth of our experiments, the value of our local cluster-wise contrastive learning, and the unified framework. Below, we respond to your questions in detail.
>
> **1. Choice of the pre-trained language model (mT5)**
>
> We adopt mT5 Base because it offers a balanced combination of semantic coverage, multilingual generality, and computational efficiency. Larger variants such as mT5 Large or XL were considered, but the base model offers a more stable and feasible trade-off between representational capacity and training cost, especially under SLU datasets that are substantially smaller than large text corpora available for conventional machine translation. Sigma remains compatible with larger models, since the backbone can be replaced directly without modifying the training objectives or the cross-modal mechanisms.
>
> **2. Computational and memory costs of SignEF and HAL**
>
> Table. Training and inference cost on WLASL2000.
> |Method|Giga-FLOPs|Peak GPU memory (GB)|
> |-|-:|-:|
> |**During training**|
> |Pre-training|
> |Sigma| 395.73 | 39.04 |
> |- SignEF| 379.23 | 38.64 |
> |- Local loss in HAL | 390.77 | 39.02 |
> |- Global loss in HAL| 393.73 | 39.03 |
> |Fine-tuning|
> | Sigma| 372.92 | 23.55 |
> |**During inference**|
> |Pre-training|
> |Sigma| 395.73 | 5.54 |
> |- SignEF| 379.23 | 5.35 |
> |- Local loss in HAL | 390.77 | 5.24 |
> |- Global loss in HAL| 393.73 | 5.28 |
> |Fine-tuning|
> |Sigma| 363.27 | 4.47 |
>
> We illustrate the computational footprint of Sigma in the above table by measuring FLOPs and GPU memory usage across both pre-training and fine-tuning, during training and inference. "-" indicates experiments conducted without the corresponding component. In practice, the system remains lightweight. SignEF introduces little additional cost, since it reuses the existing attention pathways rather than adding new fusion blocks. Both local and global loss computation in HAL contribute very little overhead, amounting to a small amount of increase in FLOPs and less than 1 GB of extra GPU memory during pre-training, where these costs only occur. They do not affect downstream fine-tuning or inference, where the computational path remains as efficient as standard mT5-based models.
>
> **3. Scalability and multilingual capability**
>
> Sigma is designed with scalability and multilingual capability in mind. Its modular structure allows components to operate independently of new models, which means the framework can grow naturally with larger skeleton corpora or stronger backbones without requiring changes to the overall training pipeline. At the same time, Sigma benefits from the multilingual generalisation ability of mT5, enabling sign-text pairs from different linguistic families, such as CSL and ASL, to be represented within a shared semantic space.
>
> **4. On excluding RGB information**
>
> Thank you for raising this important point. Our focus on skeletal data stems from its robustness to background variation, compact storage requirements, and privacy-preserving nature, which together make it a practical foundation for large-scale SLU pre-training. Nonetheless, we agree that RGB information can provide complementary cues that may be difficult to recover from skeletons alone. RGB becomes valuable when contextual scene information contributes directly to meaning, such as live presentations accompanied by slides, weather broadcasts with animated backgrounds, interactions with objects, or subtle hand-object affordances. However, such context-dependent scenarios are rare in existing SLU datasets. Our focus on skeletal data, therefore, reflects the current data landscape and focuses directly on the expressive poses themselves. We would like to explore RGB integration when more computational resources become available, although it is beyond the scope of the current work.
>
> **5. Style and formatting consistency**
>
> We appreciate this suggestion. We have revised the manuscript to ensure consistent typography. All method names (such as SignEF, and HAL) are now in a consistent style, and dataset names (such as WLASL2000) have been formatted consistently in table headers.

---

### Author Response · Authors · 2025-12-03

Dear ACs and Reviewers,

We sincerely thank everyone for their selfless dedication, constructive feedback, careful reading, and detailed suggestions. The feedback was invaluable in helping us refine the clarity, motivation, and presentation of Sigma. In response, we have improved by strengthening key explanations, adding new analyses, expanding ablations, and resolving most of the concerns regarding contribution, formulation, and empirical validation.

We note that the reviewers responded positively to several core contributions of our work. Among the four reviewers, three indicated that the paper is suitable for acceptance, and another reviewer did not catch the chance to update their assessment before the discussion phase closed. We take a positive view of this situation, as the majority of reviews already highlight Sigma’s novlety, technical soundness, clear motivation, and strong empirical results. One reviewer assigned a score of 8 and reaffirmed this assessment after reading our rebuttal, highlighting the clarity of our presentation, the soundness of our methodology, and the significance of integrating linguistic structure into skeleton-based sign language understanding (SLU). Several reviewers also recognised the architectural coherence of Sigma, the novelty of sign-aware early fusion (SignEF) and hierarchical alignment learning (HAL) within the SLU domain, and the strong empirical performance observed across diverse benchmarks. We further highlighted that the dual objective of the sign-grounded text (SGT) encoder, combining matching and generation, enables Sigma to support both recognition-oriented tasks (ISLR and CSLR) and generation-oriented tasks (SLT). This unified formulation allows the encoder to be reused seamlessly as the SGT decoder for all downstream tasks. Reviewers were no longer able to provide additional replies or adjust their scores, leaving these favourable assessments as the final record. Taken together, the feedback confirms that Sigma addresses important gaps in current SLU research and offers a meaningful and well-validated contribution to the community.

*“The paper **tackles an important limitation** of recent work in SLR and SLT, that is: that most models incorporate only visual cues and neglect the linguistic aspects of the task at hand.”* (Reviewer EVSg)

 *“The proposed components demonstrate architectural novelty in **solving clearly articulated problems** in SLU. ”* (Reviewer Kc3q)

*“Sigma **delivers consistent SOTA improvements across both recognition and translation benchmarks**, demonstrating that the framework is technically sound and performs reliably in practice.”* (Reviewer LWyo)

*“The component design of Sigma is **particularly novel**, which provides **a creative perspective on integrating skeletal information** with linguistic context”* (Reviewer q949)

---

> ### Author Response · Authors · 2025-12-03
>
> Below, we summarise the major revisions and clarifications incorporated into our work.
>
> **1. Clarified and strengthened contribution**
>
> Sigma differs from prior skeleton-based SLU models by providing a unified solution for ISLR, CSLR, and SLT, which are often treated as separate tasks with disconnected objectives. This fragmentation limits knowledge sharing across tasks and weakens semantic grounding. Sigma instead unifies them within one framework and strengthens grounding at multiple levels. By inheriting a pre-trained machine translation backbone (mT5), the model gains access to rich linguistic structure that sign-only datasets cannot provide. SignEF brings this structure into the sign encoder from the outset, guiding raw kinematic signals toward meaningful semantics rather than relying solely on shallow contrastive cues. HAL further refines this process, combining global alignment with cluster-wise interactions that approximate gloss-like units without requiring gloss annotation. Together with an SGT encoder that supports both recognition and generation, Sigma forms a coherent system that leverages linguistic priors and structured alignment to improve skeleton-based SLU in ways that conventional task-specific pipelines cannot.
>
> **2. Semantic grounding evidence**
>
> Reviewers asked for concrete evidence that Sigma improves semantic grounding. In response, we added a new analysis using Centered Kernel Alignment (CKA), which shows a consistent improvement of around 20 points in sign-text representational similarity (approximately 46 percent to 66-67 percent). We additionally show that stronger CKA correlates with improvements in BLEU, reinforcing that the enhanced grounding contributes directly to downstream task performance.
>
> **3. Clarity of cluster-wise contrastive learning**
>
> We updated the equation for HAL with explicit definitions, and clarified the roles of projection heads and global versus local similarity. The algorithm for cluster-wise similarity has been corrected to resolve concerns regarding the softmax operation and summation dimensions. We also added a new visualisation (Figure 9) and clarified that the cluster-wise similarity does not have explicit supervision but functions as a self-supervised alignment constraint.
>
> **4. Sub-word tokenisation**
>
> We clarified that Sigma does not design or choose sub-word segmentation; it follows the tokenizer built into mT5. To handle the lack of linguistic meaning in fragments like “curios” and “ity,” HAL’s cluster aggregator groups sub-words into coherent semantic clusters that serve as gloss-like alignment units. This reduces noise and improves sign-text matching.
>
> **5. Temperature scaling**
>
> We provided a clearer explanation for omitting temperature scaling in the local contrastive branch. Local alignment operates over high-variance, fine-grained sign features where temperature scaling would undesirably flatten similarity distributions and weaken sensitivity to precise correspondences. By contrast, global alignment operates over stable sentence-level features where temperature improves optimisation.
>
> **6. Module Effectiveness**
>
> We provided more explicit ablations by linking the additional experiments in Appendix L directly to the main paper. We ablate the use of our sub-modules for the unified method. These results, included in the revision, demonstrate that each module is necessary for achieving strong performance across the mainstream SLU tasks.
>
> **7. Unified Inference for ISLR, CSLR, and SLT**
>
> In terms of how the unified model performs inference, we added a clear description in Section 3.3. All tasks are treated as conditional language modelling, meaning the mT5 decoder remains active for every task. ISLR produces one or a few textual label tokens, CSLR produces gloss sequences, and SLT produces sentences, all through the same autoregressive decoding pathway. No task uses a separate classifier head, and the decoder is never removed.
>
> **8. Additional Improvements**
>
> Finally, we refined typography and formatting, expanded related works in Appendix B, added discussion on scalability and multilingual capability, and reported computational overhead for SignEF and HAL (FLOPs and memory).

---

### Meta-Review · Area_Chair_xV6S · 2026-01-02

**Summary:**

While the paper proposes a unified pre-training framework for skeleton-based sign language understanding and reports strong empirical results, reviewers raised several substantive weaknesses that limit confidence in the contribution. A critical concern raised by AC is the lack of clarity and transparency regarding the pre-training data: the paper does not clearly specify the composition, scale, or provenance of the pre-training datasets, making it difficult to assess reproducibility, fairness of comparisons, and the true source of the reported gains.
AC tried to find this detail in the paper and failed to locate such details. In other words, AC worries that whether similar data have been used in pretraining stage? As SignBert and its variants only use sign skeletons for pretraining, it can scale easily, e.g. even using uncaptioned sign videos. However, this method relies on sign-text pairs for pretraining. This need the authors to specify what data has been employed for pretraining.

Reviewers also raised concerns on the methodological novelty. In particular, early fusion via cross-attention, hierarchical/cluster-wise contrastive alignment, and joint matching-generation objectives, have been largely adapted from existing ideas rather than introducing fundamentally new techniques. In addition, despite claims of improved semantic grounding, the qualitative visualizations (e.g., Figures 10 and 11) were found to be confusing, as they do not clearly demonstrate meaningful or interpretable alignment between sign skeletons and linguistic units. Several reviewers further remarked that performance improvements are incremental, with limited insight into why the method works, and that some design choices (e.g., subword-level alignment, omission of temperature scaling) remain insufficiently justified despite added explanations.

**Reviewer Concerns:**

**Concerns that were addressed by the rebuttal:**

The rebuttal did make a genuine effort to respond to several technical questions raised by the reviewers. In particular, the authors clarified the overall architecture and motivation of the framework, added more ablation studies to show the contribution of individual components, and provided additional quantitative analysis (e.g., CKA scores) to support the claim of improved semantic grounding. Some presentation issues were also cleaned up, including clearer explanations of the hierarchical alignment mechanism, temperature scaling choices, and the unified inference formulation across ISLR, CSLR, and SLT. These responses helped address how the method is constructed and trained.

**Concerns that remain outstanding:**

However, several key concerns remain unresolved. Most importantly, the pre-training dataset is still not clearly specified: the paper does not adequately describe what data is used for pre-training, how large it is, or how it differs from downstream datasets. This missing information makes it difficult to assess reproducibility and to understand whether the reported gains come from the proposed method or from data advantages. Another interpretation of the pretraining is the paper only uses the target dataset for pretraining. However, this should not be considered as pretraining. This makes the whole paper very confusing.

In addition, the qualitative evidence for alignment remains weak. Figures 10 and 11 are intended to demonstrate semantic or sign–text alignment, but the visualizations are confusing and do not clearly show meaningful correspondence, which undermines claims of interpretability and grounding. Moreover, why not comparing sign-text alignment of models that have been trained? Only showing pretraining model closes the gap is not a strong argument.

More broadly, concerns about limited methodological novelty have been raised by several reviewers since the core components largely adapt existing techniques rather than introducing fundamentally new ideas.

**Reviewer Scores:**

This paper received mixed feedback, with two positive and two negative reviews. After carefully reading the rebuttal, I do not find the response sufficiently convincing to resolve the key concerns raised by the negative reviewers. In particular, the explanation of why early fusion leads to better sign–text feature alignment remains largely intuitive and descriptive, rather than being supported by strong empirical or analytical evidence.

While the results suggest that pretraining helps bridge the sign–text gap, it is still unclear why the proposed pretraining strategy, i.e., the early fusion, is superior to existing pretraining or late-fusion approaches. The comparisons and alignment analyses provided are not strong enough to clearly isolate the benefit of early fusion itself. As a result, the core question of what enables the observed gains remains insufficiently answered. Furthermore, the subword clustering is not clearly stated. This is purely text operation, why it help sign-text alignment? Figures 5 and 6 do not show the confidence of grounding. It is more or less like an illustration rather than output of a model.
Given these unresolved issues, I believe it is unlikely that the reviewers who initially expressed concerns will substantially revise their scores upward.

---

### Decision · Program_Chairs · 2026-01-26

Reject